# The Impact of Development Zones on China's Urbanization from the Perspectives of the Population, Land, and the Economy

**Kaimeng Li [1,†], Shuang Gao [2,†], Yuantao Liao [1], Ke Luo [1] and Shaojian Wang [2,\*]**

[1] Guangzhou Urban Planning & Design Survey Research Institute, Guangzhou 510060, China
[2] Guangdong Provincial Key Laboratory of Urbanization and Geo-Simulation, School of Geography and Planning, Sun Yat-sen University, Guangzhou 510275, China
[\*] Correspondence: wangshj8@mail.sysu.edu.cn
[†] These authors contributed equally to this work.

**Abstract:** The sustainable development of urbanization is a necessary condition for China to realize modernization. Considering the importance of urbanization to China's future development and the advantages of development zones in promoting urbanization, it is necessary to quantify the impact of establishing development zones on urbanization development. Using the difference in difference (DID) model, this study takes the panel data of 235 cities in China from 1990 to 2017 to evaluate the policy effects of setting up development zones on urbanization from the perspectives of the population, land, and the economy. The results show that the development zone policy in the overall panel exerts a significant negative impact on land urbanization and a significant positive impact on economic urbanization but exerts no significant impact on population urbanization. The regression results of sub-regions show significant regional differences in the impact of development zones on urbanization. In the eastern region, the development zone policy has promoted the intensive use of urban construction land. For the central and western regions with weak development foundations, development zones play an important role in attracting the population and upgrading industries while reducing the intensive use of construction land. This study provides urban-level empirical evidence for evaluating the urbanization effects of development zone policies and puts forward policy recommendations for development zone construction to promote high-quality urbanization in China.

**Keywords:** development zone; urbanization; policy effect; difference in difference model; China

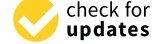



## 1. Introduction

The emergence of cities is a sign of human maturity and civilization [1]. After two centuries of unprecedented rapid urbanization, more than half of the world's population lives in cities [2]. Rapid urbanization has a profound impact on global sustainability while increasing social productivity. The research on urbanization began with the emergence of modern industrial cities and has drawn widespread concern from the academic community. Scholars believe that while urbanization promotes social modernization in various countries, it also brings a series of problems. For example, studies in London and New York found that state capital and planning play an essential role in urbanization [3]. In urban renewal, land value and rents have risen rapidly. However, ordinary citizens do not benefit from this, and the wealth gap problem becomes more significant through urbanization [4]. The rapid development of urbanization has also deepened environmental issues and aroused academic attention regarding healthy and ecological cities [5]. Scholars have deeply analyzed urban air pollution [6], the heat island effect [7], ecological patterns [8], land expansion [9], and other issues under the process of urbanization, and explored sustainable urbanization models.

Since 1990, China's urbanization has entered a stage of rapid development [10]. Urbanization not only drives large-scale population agglomeration to cities [11], but also

promotes urban spatial and industrial restructuring [12]. Sustainable urbanization is a vital driving force and is necessary for national modernization [13,14]. For China, guiding sustainable urbanization not only determines the future of China's urbanization but also affects the prospects of world urbanization [15]. Therefore, scholars have deeply analyzed the influencing mechanism of urbanization and explored sustainable urbanization models. Some scholars have summarized China's urbanization development into seven "promoting models". Among these, the "development zone (DZ) model" is one of the most representative [16]. The "DZ model" is a typical example of government-led urbanization. The government establishes DZ in regions with advantageous locations and provides a series of preferential policies [17]. DZs can accept the transfer of international capital and industries by their policy superiority and favorable investment environment [18]. By establishing DZs, cities can complete industrial and population agglomeration [19], and achieve the transformation of the land and industrial structures [20]. In addition, the goal of DZs has gradually changed from economic output to industry–city integration [21]. Since the establishment of the first national DZ in 1984, DZs have become an important strategic initiative to promote local development [22]. As of 2018, China has established a total of 552 national DZs. DZs have also become a significant way in which to promote the sustainable urbanization of China.

The coordinated development of the urban population, economy, and land use is important to reflect the sustainability of urbanization. Therefore, judging whether a DZ can significantly drive the transformation of the urban population, industry, and land use provides an indispensable window through which to explore the sustainable development model of China's urbanization. Previous studies have proved that establishing DZs can improve cities' economic level and production efficiency [23]. However, there are also some problems in the development process of DZs, which lead to insufficient population agglomeration and even industrial hollowing [24]. As an essential spatial carrier of urbanization, can the DZ policy promote the coordinated development of the urban population, industry, and land use? This has become the core issue of sustainable urbanization for the future.

Some scholars have realized that DZs exert a promoting effect on urbanization. However, thus far, no research has explored the impact of DZ policies on urbanization from the perspectives of the population, land, and economy. Through in-depth research on the process of urbanization in different disciplines, the concept of urbanization has also been extended [25]. Urbanization has gradually become a comprehensive process involving population migration, industrial structure adjustment, and land-use pattern transformation [26]. However, few studies have explored the urbanization development model from a comprehensive perspective. To fill the above research gaps, this study uses the DID model to quantify the role of DZs in China's urbanization. We conduct a series of robustness analyses to exclude the influence of external circumstances on the results. Considering that urbanization is a comprehensive process, we evaluate the spillover effects of DZs on urbanization from the perspectives of the population, land, and the economy. In addition, this study identifies the heterogeneous influence of DZs on urbanization to serve targeted urbanization development policies.

## 2. Literature Review

The industrial agglomeration policy represented by DZs is an essential means for the country to promote regional development [27]. Benefiting preferential policies and financial subsidies, DZs can quickly achieve industrial agglomeration, thus driving regional development [28]. Since the 17th century, DZs have been widely set up in various countries and regions worldwide, becoming the most popular industrial policy in both developed and developing countries [29,30]. The policy effect of DZs has also become the focus of urban economics and urban planning.

A large number of studies have focused on exploring the economic effects of DZs from the perspectives of regions and firms. Empirical research shows that DZs are key in improving regional economic conditions [31]. Attracted by preferential policies, capital and

enterprises enter a DZ to conduct business [32]. By setting up DZs, cities attract investment to develop industries and enhance the vitality of economic activities. Industrial development also provides opportunities to improve urban employment and income [33,34]. Many DZs have become the growth poles of the regional economy. In addition, scholars have analyzed the impact of DZs on local economic development from the perspective of enterprises and also obtained positive conclusions. It has been found that the establishment of a DZ dramatically improves the economic output of the enterprises in the zone, especially capital-intensive enterprises [35]. The economic agglomeration effect brought about by the DZ not only improves the production efficiency of its internal enterprises, but also promotes the production efficiency of its surrounding enterprises. However, although many cases confirm the positive impact of DZs on economic development, not all of them are successful. The actual role that some DZs play is small or even far less than their investment [36]. For example, a study in India has shown that establishing DZs has not promoted local socio-economic development [24]. The success of the DZ practice in Britain has also not been confirmed. Although DZs in Britain undertake a large number of economic activities, policies based on DZs do not play a prominent role in promoting regional economic growth [37]. Faced with the different impacts of DZs, scholars have concluded that DZs cannot be regarded as effective catalysts for all countries [38].

In China, many studies have confirmed the positive effects of the DZ policy and the promotion of urban development through the following aspects. First, the establishment of the DZ is accompanied by large-scale infrastructure construction, which improves the spatial appearance of the city [39]. Second, the DZ attracts foreign investment into the city through preferential policies, forming industrial agglomerations and becoming the growth pole of the city [40]. Third, enterprises settled in the DZ increase urban tax revenue and attract labor accumulation [35]. Nevertheless, the development of China's DZs has not always been smooth. On the one hand, due to the excessive proportion of foreign investment in DZs, they are affected by the international economic cycle. On the other hand, the abuse of preferential policies also leads to the spillover effects of DZs no longer being obvious [41]. Between 1990 and 2005, due to the blind establishment of DZs by local governments, the phenomenon of "zone fever" frequently occurred in China [42]. This resulted in the positive effects of DZs no longer being significant [43].

Unfortunately, few studies have focused on the impact of DZs on urbanization, especially the exploration of sustainable urbanization models under the influence of DZs. Nevertheless, the research on the relationship between DZs and urban land use and the labor market provides us with new perspectives. The development of a DZ is accompanied by large-scale construction activities, including the construction of infrastructure, an industrial zone, and support for living service facilities [44]. DZs influence the layout of urban space, such as by accelerating the expansion of urban spaces and changing urban spatial structure [45]. Therefore, scholars generally believe that the establishment of DZs promotes the process of urban suburbanization and urbanization. In addition to land use, the impact of DZs on the labor market has also attracted scholars' attention. Most studies on the labor market of DZs in the United States have not yielded positive conclusions [46,47]. However, research on DZ policies in European countries has found that the construction of DZs can attract large numbers of the employed population [48]. DZs have created economic and substantial social benefits, and are gradually becoming the primary sector to absorb the employed population [49]. We believe that DZs play a key role in promoting urban population agglomeration.

The policy effect of DZs has attracted the attention of a large number of scholars, and their research results have very crucial reference value for this study. However, existing studies do not exclude the external environment's influence when analyzing the policy effects of DZs. Thus, the policy effects of DZs on urbanization have not been thoroughly examined. Existing studies only consider the impact of DZs on a single index, ignoring the comprehensive nature of urbanization. Judging whether a DZ can coordinate the transformation of the urban population, industry, and land use is of great significance when

exploring the sustainable development mode of urbanization in China. In addition, the spatial differences in the policy effects of DZs have not been fully considered.

## 3. Data and Methodology

### 3.1. Difference in Difference (DID) Model

This study aims to test whether the development zone policy can promote the urbanization process. The difference in difference (DID) model is practical for evaluating policy effects. The model divides the research sample into a treatment group (in which the policy is implemented) and a control group (in which the policy is not implemented). Unobservable factors are eliminated by differentiating between before and after policy implementation and between the treatment group and the control group, so that the policy net effect can be identified. Scholars usually use the DID model to conduct policy evaluation because this model can better avoid the endogenous effects of policies [50–52]. This study divides the research sample into a treatment group and a control group according to whether the city has already established a DZ. This allows the DID model to be constructed based on the double fixed effects of time and the individual. As the time taken to establish DZs in various cities is not consistent, this article chooses the progressive DID model to test the impact of DZs on urbanization [53,54]. The expression of the model is as follows:

$$lnU_{it} = \alpha_0 + \alpha_1 DZ_{it} + \alpha_2 lnZ_{it} + \delta_t + \gamma_i + \varepsilon_{it}, \tag{1}$$

where $U_{it}$ is the dependent variable, which represents the population urbanization (PU), land urbanization (LU), and economic urbanization (EU) levels of city $i$ at time $t$, respectively; $DZ_{it}$ is the independent variable, which indicates whether city $i$ has set up a DZ at time $t$; $Z_{it}$ represents the control variables such as per capita GDP (PcGDP), average wage (AS), budgetary revenue (BR), fixed asset investment (FAI), foreign direct investment (FDI), political status (PS), nearshore distance (ND), and latitude (LA); $\alpha_0$, $\alpha_1$, and $\alpha_2$ are regression coefficients; $\delta_t$ and $\gamma_i$ are the fixed effects on the time and individual, respectively; and $\varepsilon_{it}$ is a random disturbance term. In order to eliminate the possible heteroscedasticity in the model, the dependent variables and the control variable are processed logarithmically in this study. In the model regression, the coefficient $\alpha_1$ is the focus of this study. The coefficient reflects the impact of the DZ policy on the urbanization process after double difference. If $\alpha_1$ is significantly positive, it means that the DZ policy can promote the regional urbanization process.

The DID model requires that the division of the treatment group and the control group is random. Additionally, the treatment group and the control group should also meet the parallel trend assumption before the implementation of the policy. Therefore, this study conducts propensity score matching analysis and a parallel trend test on the sample to ensure the accuracy of the estimation results. In addition, this study divides the sample into the eastern, central, and western regions to identify the differences among policy effects in different regions of China.

### 3.2. Variables and Data

Using the DID model, this study takes the panel data of 235 Chinese cities from 1990 to 2017 as the research sample and evaluates the spillover effects of DZ on urbanization. The independent variable of this research is DZ policy, which is a dummy variable. If the city did not set up a DZ during the study period, the value for all years is 0. If the city sets up a DZ during the study period, the value is 0 before the construction year and 1 after the construction year. If the city sets up multiple DZs during the study period, the construction year of the first DZ shall be regarded as the policy implementation year. The DZ data come from the "2018 Edition of the China Development Zone Directory". It is worth mentioning that the research objects of this study are national DZs, which exclude provincial and municipal DZs.

Urbanization is not only manifested in the transfer of the rural population to the urban population, but is also accompanied by the adjustment of the industrial structure and

the spread of construction land. The development process of urbanization involves the population, land, economy, and other fields [55,56]. When exploring the spillover effects of DZ policy, the dependent variables comprehensively consider population urbanization (PU), land urbanization (LU), and economic urbanization (EU). PU is the ratio of the urban population to the total population of a city. LU is the ratio of urban construction land area to the total area. EU is the proportion of the output value of the city's secondary and tertiary industries to its GDP. The population data come from the "China Regional Statistical Yearbook 1990–2018". The urban construction land data come from the Environmental Science Data Center of the Chinese Academy of Sciences, and the economic data come from the "China City Statistical Yearbook 1990–2018".

The development of urbanization is not only affected by the DZ policy, but also by socioeconomic and natural factors. Referring to existing studies [57,58], we select the city's per capita GDP (PcGDP), average wage (AS), budgetary revenue (BR), fixed asset investment (FAI), foreign direct investment (FDI), political status (PS), nearshore distance (ND), and latitude (LA) as control variables. PcGDP and AS reflect the social and economic development level of the city. PS reflects the development prospects of the city [59]. The control variables BR, FAI, and FDI are closely related to the level of urban infrastructure. These socioeconomic variables may impact urbanization development. Furthermore, the ND and LA of cities can affect the landforms, climate, temperature, and other natural characteristics of Chinese cities [60,61]. For example, as the distance from the coastline increases, the topography of Chinese cities changes from plains to mountains, and the climate changes from humid to arid. With increasing latitude, the temperature of Chinese cities changes from high to low. These natural conditions also exert an important impact on the urbanization process. Data on PcGDP, AS, BR, FAI, and FDI are from the "China City Statistical Yearbook 1990–2018". PS is a dummy variable, set as equal to 1 if the city is a municipality directly under the central government, a sub-provincial city, or a city under separate state planning; otherwise, PS is set to 0. The ND is derived from spatial statistics and is characterized by the closest distance to the coastline from the city government. LA is the spatial latitude of the city government. Table 1 reports the definition and descriptive statistics of the main variables in this article.

**Table 1.** Definition and descriptive statistics of variables.

| Variable | Definition | Obs | Mean | S.D. |
|---|---|---|---|---|
| DZ | Development zone | 6580 | 0.34 | 0.47 |
| PU | Population urbanization (%) | 6580 | 61.21 | 24.13 |
| LU | Land urbanization (%) | 6580 | 2.69 | 4.57 |
| EU | Economic urbanization (%) | 6580 | 82.17 | 12.38 |
| PcGDP | Per capita GDP (CNY) | 6580 | 23067 | 36134 |
| AS | Average salary (CNY) | 6580 | 20622 | 19253 |
| BR | Budgetary revenue ($10^8$ CNY) | 6580 | 65.19 | 163.06 |
| FAI | Fixed asset investment ($10^8$ CNY) | 6580 | 548.44 | 1046.66 |
| FDI | Foreign direct investment ($10^4$ USD) | 6580 | 33963 | 90508 |
| PS | Political status | 6580 | 0.12 | 0.33 |
| ND | Nearshore distance (km) | 6580 | 434.54 | 401.54 |
| LA | Latitude | 6580 | 32.94 | 6.73 |

## 4. Results

### 4.1. Evolution Trend of Urbanization

Table 2 shows the statistical results of urbanization indicators in the main years of the treatment and control groups. It can be seen from Table 2 that the PU, LU, and EU indicators showed a clear upward trend for both treatment and control groups. This indicates that China's urbanization process was significant. Overall, the average urbanization indicators of cities in the treatment group were significantly higher than those in the control group in all years. Taking 2017 as an example, the level of PU in the treatment group was 69.4%,

which was 1.08 times that of the control group. The levels of the LU and EU were 6.23% and 95.2%, which were 1.27 times and 1.02 times that of the control group, respectively. The difference in the LU index between the treatment group and the control group was the most significant, while the EU index was the closest. In terms of growth rate, the average annual growth rate of PU in the treatment group was 0.71%, which was lower than the 0.89% in the control group. However, the average annual growth rates of LU and EU in the treatment group cities were 5.84% and 0.99%, both higher than those in the control group cities.

**Table 2.** Urbanization evolution trend of treatment group and control group.

| Year | PU (%) | | LU (%) | | EU (%) | |
|---|---|---|---|---|---|---|
| | Treatment | Control | Treatment | Control | Treatment | Control |
| 1990 | 57.0 | 49.2 | 1.27 | 1.01 | 72.3 | 68.2 |
| 1995 | 60.3 | 53.1 | 1.58 | 1.45 | 78.5 | 75.3 |
| 2000 | 62.4 | 55.6 | 1.93 | 1.88 | 81.8 | 77.2 |
| 2005 | 67.0 | 61.0 | 2.54 | 2.46 | 85.6 | 81.4 |
| 2010 | 68.2 | 63.1 | 3.57 | 3.26 | 89.0 | 85.6 |
| 2017 | 69.4 | 64.1 | 6.23 | 4.91 | 95.2 | 93.1 |

Figure 1 displays the spatial pattern of the PU, LU, and EU levels for the treatment and control groups in 1990 and 2017. From Figure 1, the PU, LU, and EU levels varied significantly among cities. The difference in the LU level between cities was the most significant, followed by PU. Additionally, the EU level was relatively balanced among cities. These reflect the complexity of the urbanization process. Therefore, it was necessary to measure the impact of DZs on urbanization from different perspectives. Overall, the urbanization level in the eastern coastal regions was observably higher than that in the central and western regions. The Beijing–Tianjin–Hebei Urban Agglomerations, Yangtze River Delta Urban Agglomerations, and Pearl River Delta Urban Agglomerations were the high-value areas in terms of the urbanization level, and the increase was the most obvious during the study period. The regional difference in the urbanization level proved that it was reasonable to divide the sample into three regions. In addition, Figure 1 further confirms that the urbanization level of the treatment group cities was higher than that of the control group cities.

The results of the statistical analysis show that the urbanization level of cities with DZs was significantly higher than that of cities without DZs. We cannot determine that the urbanization advantage of the treatment group comes from the DZ policy, as the endogeneity of the selection of the treatment group and the control group was not resolved. The state is more inclined to set up DZs in cities with better development foundations and prospects to maximize policy benefits. Therefore, to avoid the problem of policy endogeneity affecting the model results, we performed a propensity score matching analysis on the research sample.

*4.2. Propensity Score Matching (PSM) Analysis*

When performing the DID model regression, it is necessary to ensure that the division of the treatment group and the control group is random. Therefore, before establishing a DZ, the treatment group and the control group should be as similar as possible in urban development conditions, except for the urbanization level [62]. This article used the kernel matching method in the logit model to solve the possible endogeneity problem of the policy [63]. Based on the five observed variables of PcGDP, AS, BR, FAI, and FDI, propensity score matching was carried out to determine the treatment group and the control group. Additionally, the differences between the treatment group and the control group before the establishment of DZs were reduced. The steps for propensity score matching were as follows: (1) using the Logit regression model to measure the propensity scores of the observed variables in the treatment group and the control group; (2) matching the observed variables so that the standardized deviation was controlled to within 10%;

(3) calculating the difference in the observed variables of the cities in the treatment group before and after the establishment of the DZ, and calculating the difference in the observed variables of the cities in the control group that matched them; (4) checking the balance of the samples before and after matching. The results show that there were no significant differences in the observed variables between the treatment group and the control group after matching (Table 3), thereby indicating that the matching was effective [64]. Finally, based on the successfully matched samples, the DID method was used to test the net effect of the DZ policy.

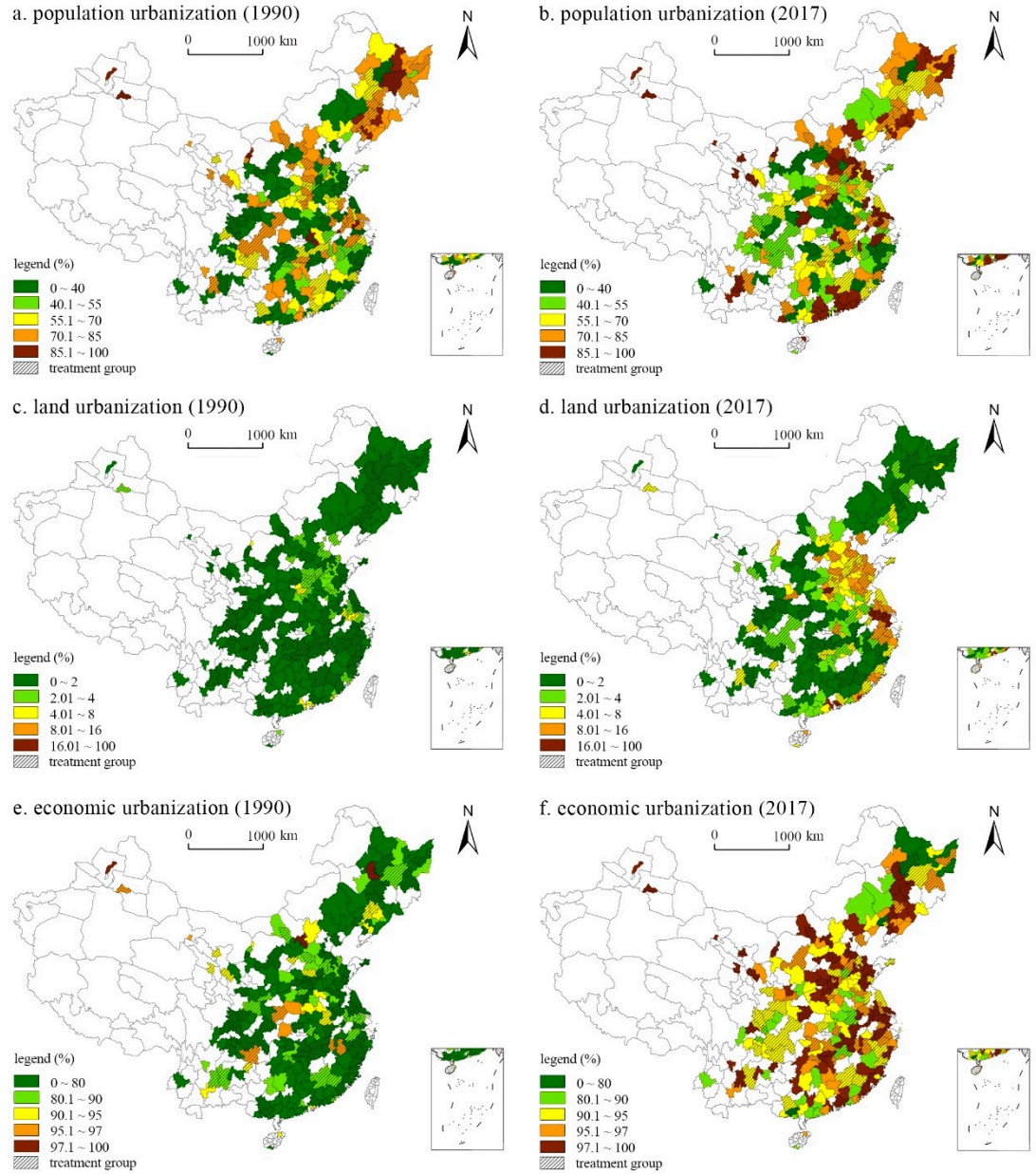

**Figure 1.** Urbanization spatial pattern of treatment group and control group in China.

**Table 3.** Results of the balance test.

| Variable | Before Matching | | | After Matching | | |
|----------|-----------------|--|--|----------------|--|--|
| | **Treatment** | **Control** | **Difference** | **Treatment** | **Control** | **Difference** |
| PcGDP | 17,018 | 6938 | 66.5 *** | 12,405 | 11,541 | 5.7 |
| AS | 9773 | 7600 | 90.2 *** | 9182 | 8718 | 19.3 |
| BR | 24.98 | 4.74 | 86.5 *** | 12.48 | 12.86 | −1.6 |
| FAI | 143.03 | 46.13 | 101.9 *** | 94.03 | 80.50 | 14.2 |
| FDI | 33,408 | 5527 | 75.9 *** | 20,846 | 19,596 | 3.4 |
| N | 56 | 179 | - | 38 | 151 | - |

Note: *** indicate significance at the 1% level.

### 4.3. Parallel Trend Test

The DID model requires the treatment group and the control group to pass a parallel trend test to ensure the accuracy of the estimated results [53]. The principle of the parallel trend test was to introduce the dummy variable into the regression to measure the year-by-year impact of the d DZ policy. The model expressions were as follows:

$$lnU_{it} = \alpha_0 + \alpha_1 DZ_{it}^{-5} + \alpha_2 DZ_{it}^{-4} + \cdots + \alpha_{10} DZ_{it}^{4} + +\alpha_{11} DZ_{it}^{5} + \delta_t + \gamma_i + \varepsilon_{it}, \quad (2)$$

where $U_{it}$ is the dependent variable, which represents the population urbanization (PU), land urbanization (LU), and economic urbanization (EU) levels of city $i$ at time $t$, respectively; $DZ_{it}$ is the dummy variable for establishing the DZ; $DZ_{it}^{-5}$ indicates that time $t$ is 5 years before the establishment of the DZ in city $i$, it is set as one; otherwise, it is set as zero; $DZ_{it}^{5}$ indicates that if time $t$ is 5 years after the establishment of the DZ in city $i$, it is set as one; otherwise, it is set as zero; and other dummy variables are defined similarly; $\delta_t$ and $\gamma_i$ are the fixed effects on time and the individual, respectively; and $\varepsilon_{it}$ is a random disturbance term. Based on the regression results of the above model (95% confidence level), we drew a parallel trend test plot, as shown in Figure 2, adjusted for city clustering.

As can be seen from Figure 2, the establishment of DZs exerted an impact on urbanization. Additionally, the impact of DZs on urbanization increases significantly over time. The policy effect on urbanization in years without DZs fluctuated around the zero value. This indicated that there were no unparallel trends in the sample before the city set up the DZ. However, after the establishment of DZs, the policy effect of PU, LU, and EU significantly deviated from a zero value. This shows that the establishment of DZs exerted an impact on urbanization progress. Additionally, the treatment and control groups in this study also passed the parallel trend test. The establishment of DZs in cities significantly affected the inherent processes of PU, LU, and EU. Specifically, the DZ policy exerted an incredibly positive effect on PU and EU, while it negatively affected LU. In addition, Figure 2 showed that the policy effect of DZs on urbanization increases significantly over time. Due to this, we have reason to believe that establishing DZs can significantly change the developmental trend of the urban population, land use, and industrial industry, and these impacts will become more pronounced over time.

### 4.4. The DID Model Regression

With the help of the DID model, this study explores the impact of the establishment of DZs on the urbanization level of cities from the three aspects of population, land, and the economy, and controls the time and individual fixed effects. The results are shown in Table 4. There were no control variables introduced in columns (1), (3), and (5). Control variables such as PcGDP, AS, BR, FAI, FDI, ND, and LA were introduced in columns (2), (4), and (6).

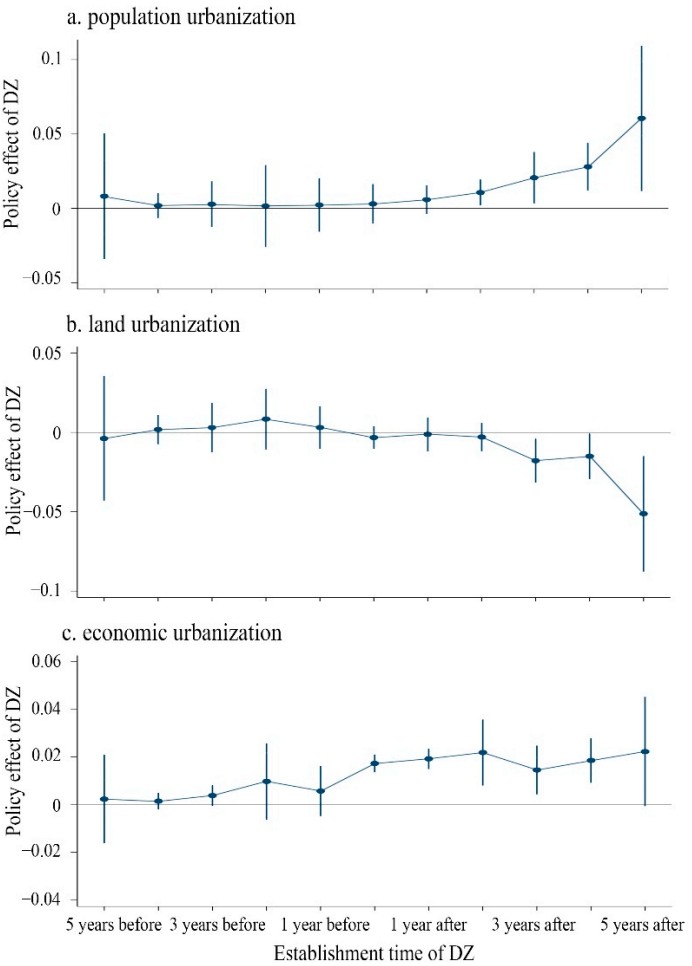

**Figure 2.** Parallel trend test plots.

Column (1) of Table 4 shows that establishing DZs improved the urban PU at the 1% significance level, with a coefficient of 0.193. After introducing the control variables, the coefficient of PU became 0.049, but it did not pass the significance test. We have no sufficient evidence to prove that establishing DZs in cities can significantly increase PU. The purpose of establishing the DZs was to develop high-tech industries and become the growth poles of the city and surrounding areas [65]. However, the industries within the DZ are mostly knowledge-intensive and technology-intensive enterprises that have a relatively low demand for an unskilled labor force. Therefore, setting up a DZ cannot attract many laborers into the city, nor can it significantly increase the urban PU level. The coefficients of the control variables show that PcGDP, AS, FDI, and LA significantly improved the urban PU level, which was consistent with the conclusions of related studies.

The results in columns (3) and (4) of Table 4 show that establishing DZs can significantly reduce urban LU, regardless of whether control variables are introduced or not. The coefficient was 0.519 when no control variables were added. After adding the control variables, the coefficient became 0.026, indicating that its degree of influence was weakened. It can be seen from Table 2 and Figure 1 that during the study period, the LU in Chinese cities showed a rapid upward trend, indicating a significant expansion of construction land. However, the results of this study support the premise that compared with cities without DZs, cities with DZs had a lower degree of expansion of construction land. We attribute this to the fact that DZs were industrial space policies established by the government in specific areas of the city, with clear boundaries. Therefore, the land use within the DZ was also more intensive and efficient. To some extent, the DZ restrained the large-scale expansion of construction land in its host city. The coefficients of PcGDP, BR, and FDI in the

control variables were significantly positive, indicating that they significantly improved the LU. It is worth noting that the coefficient of ND was significantly negative, which means that the distance of a city from the coastline can reduce its LU level. It follows that the degree of intensive land use in China's coastal cities was higher than that of inland cities.

**Table 4.** Basic regression of the impact of DZs on urbanization.

| Variable | lnPU | | lnLU | | lnEU | |
|---|---|---|---|---|---|---|
| | **(1)** | **(2)** | **(3)** | **(4)** | **(5)** | **(6)** |
| DZ | 0.193 *** | 0.049 | −0.519 *** | −0.226 * | 0.063 *** | 0.032 ** |
| | (0.048) | (0.053) | (0.163) | (0.108) | (0.013) | (0.108) |
| lnPcGDP | | 0.160 *** | | 0.476 *** | | 0.055 *** |
| | | (0.057) | | (0.099) | | (0.017) |
| lnAS | | 0.003 | | 0.073 | | 0.133 *** |
| | | (0.129) | | (0.249) | | (0.029) |
| lnBR | | 0.049 | | 0.352 *** | | 0.048 *** |
| | | (0.029) | | (0.071) | | (0.010) |
| lnFAI | | 0.021 | | 0.071 | | −0.007 |
| | | (0.026) | | (0.057) | | (0.009) |
| lnFDI | | 0.016 * | | 0.050 ** | | −0.005 ** |
| | | (0.008) | | (0.019) | | (0.002) |
| PS | | 0.196 *** | | 0.212 | | 0.037 ** |
| | | (0.068) | | (0.156) | | (0.015) |
| lnND | | 0.012 | | −0.135 *** | | 0.009 *** |
| | | (0.018) | | (0.034) | | (0.003) |
| lnLA | | 0.584 *** | | 0.074 | | 0.038 |
| | | (0.131) | | (0.305) | | (0.032) |
| _cos | 3.807 *** | 0.385 | | −3.988 *** | 4.208 | 2.603 *** |
| | (0.041) | (1.045) | | (1.938) | (0.023) | (0.223) |
| Year | Yes | Yes | Yes | Yes | Yes | Yes |
| City | Yes | Yes | Yes | Yes | Yes | Yes |
| N | 5292 | 5292 | 5292 | 5292 | 5292 | 5292 |
| $R^2$ | 0.073 | 0.240 | 0.181 | 0.466 | 0.232 | 0.423 |

Note: The values in parentheses are the standard deviations; *, **, and *** indicate significance at the 10% level, 5% level, and 1% level.

The results in columns (5) and (6) of Table 4 show that DZs significantly improved the urban EU, with a coefficient of 0.032 after the introduction of control variables. Existing studies have shown that DZs were important economic growth poles in the cities where they were located [40], as evidenced by their high proportion of GDP in these cities. Generally, secondary industries dominate DZs in order to promote the upgrading of the industrial structure of their host cities. After the establishment of DZs in cities, the proportion of secondary industry greatly increased, and the EU level also significantly improved. The regression results of the control variables show that there was a significant positive correlation between PcGDP, AS, BR, PS, ND, and EU.

*4.5. Heterogeneity Analysis*

We used the DID model to empirically analyze the impact of the establishment of DZs on urbanization and concluded that DZs could reduce LU and improve EU. However, whether this conclusion was valid in different regions of China remains to be tested. The impact of the establishment of DZs on urbanization also depends on the level of regional economic development and industrial structure. Although DZs were established under the guidance of national policies, the population attractiveness, infrastructure construction level, and economic development level of DZs in different regions varied greatly, so the impact of DZs on regional urbanization will also vary. Therefore, we divided the research samples into eastern, central, and western regions and further investigated the heterogeneity impact of DZs on urbanization. The results are shown in Tables 5–7.

**Table 5.** Regional regression of the impact of DZs on population urbanization.

| Variable | Eastern Region | | Central Region | | Western Region | |
|---|---|---|---|---|---|---|
| | (1) | (2) | (3) | (4) | (5) | (6) |
| DZ | 0.099 | 0.029 | 0.249 *** | 0.081 * | 0.323 ** | 0.081 * |
| | (0.098) | (0.101) | (0.059) | (0.057) | (0.137) | (0.097) |
| Control | No | Yes | No | Yes | No | Yes |
| Year | Yes | Yes | Yes | Yes | Yes | Yes |
| City | Yes | Yes | Yes | Yes | Yes | Yes |
| N | 1960 | 1960 | 2324 | 2324 | 1008 | 1008 |
| $R^2$ | 0.127 | 0.233 | 0.058 | 0.250 | 0.103 | 0.587 |

Note: The values in parentheses are the standard deviations; *, **, and *** indicate significance at the 10% level, 5% level, and 1% level.

**Table 6.** Regional regression of the impact of DZs on land urbanization.

| Variable | Eastern Region | | Central Region | | Western Region | |
|---|---|---|---|---|---|---|
| | (1) | (2) | (3) | (4) | (5) | (6) |
| DZ | −0.544 ** | −0.060 * | 0.340 | −0.134 | 0.605 ** | 0.043 * |
| | (0.178) | (0.203) | (0.264) | (0.153) | (0.300) | (0.192) |
| Control | No | Yes | No | Yes | No | Yes |
| Year | Yes | Yes | Yes | Yes | Yes | Yes |
| City | Yes | Yes | Yes | Yes | Yes | Yes |
| N | 1960 | 1960 | 2324 | 2324 | 1008 | 1008 |
| $R^2$ | 0.254 | 0.508 | 0.194 | 0.429 | 0.214 | 0.469 |

Note: The values in parentheses are the standard deviations; *, and ** indicate significance at the 10% level and 5% level.

**Table 7.** Regional regression of the impact of DZs on economic urbanization.

| Variable | Eastern Region | | Central Region | | Western Region | |
|---|---|---|---|---|---|---|
| | (1) | (2) | (3) | (4) | (5) | (6) |
| DZ | −0.046 ** | −0.012 * | 0.077 *** | 0.030 * | 0.061 | 0.014 |
| | (0.020) | (0.013) | (0.028) | (0.021) | (0.039) | (0.020) |
| Control | No | Yes | No | Yes | No | Yes |
| Year | Yes | Yes | Yes | Yes | Yes | Yes |
| City | Yes | Yes | Yes | Yes | Yes | Yes |
| N | 1960 | 1960 | 2324 | 2324 | 1008 | 1008 |
| $R^2$ | 0.345 | 0.565 | 0.176 | 0.334 | 0.222 | 0.600 |

Note: The values in parentheses are the standard deviations; *, **, and *** indicate significance at the 10% level, 5% level, and 1% level.

Table 5 presents the estimated results of the DZs in different regions regarding PU, from which we can conclude that there are some differences between the regional regression results and the overall national situation. For the central and western regions, the establishment of DZs had a positive impact on PU and passed the significance test. Notably, after introducing the control variables, the influence coefficients of the central region and the western region were both 0.081. The impact of DZs on PU did not significantly differ between the central and western regions. For the eastern region, the impact of DZs on PU did not pass the significance test. A possible reason for this was that the eastern region, as an economically developed region in China, had a higher level of technology in the DZ industries, so it was less dependent on the magnitude of the labor force. As for the central and western regions, their industries were often transferred from the eastern regions and were mainly labor-intensive industries. In addition, compared with the eastern region, the cities in the central and western regions were limited by their lower development level and were less attractive to the labor force. After the establishment of DZs in cities in the central

and western regions, their attractiveness to the labor force was much higher than that of surrounding cities, so PU obviously improved.

The results in Table 6 show that there were also significant spatial differences regarding the impact of establishing DZs on LU. After introducing the control variables, the regression coefficient of the DZ in the eastern region of LU was negative and significant at the 10% level. In comparison, the regression coefficient of the DZ in the western region on LU was positive and significant at the 10% level. These results indicate that the establishment of DZs limited the expansion of construction land in the eastern region, and its land use was more intensive. The establishment of DZs in cities in the western region promoted the expansion of construction land. This study attributes this differential response to the premise that after the long-term and rapid development of cities in the eastern region, the amount of construction land was relatively small and the land price was high, making their land use more efficient and intensive. In this way, establishing DZs further promoted more intensive land-use development in the eastern region. As for the western region, its construction land was in the stage of rapid expansion, and the establishment of DZs further increased the expansion speed of construction land. For the central region, the impact of DZs on LU was positive without including control variables and negative upon introducing the control variables; however, both failed to pass the significance test. The reason for this may be that the positive and negative effects of the DZ on LU in the central region cancel each other out in the whole sample.

Table 7 shows that DZs had a significant negative impact on EU in the eastern region but a significant positive impact in the central region. We attributed this result to the difference in development levels between the eastern and central regions. The eastern cities were more developed, and the proportion of the agricultural output in the GDP was lower. The impact of the establishment of DZs on EU manifested itself more in technical improvement, which was not within the scope of this study. Therefore, the establishment of DZs in eastern cities cannot increase the proportion of their industrial and service production in GDP. For cities in the central region, the proportion of agricultural production was greater than that in the eastern region. However, after the establishment of DZs, the proportion of secondary production increased significantly, thereby promoting the EU level.

## 5. Discussion

Although the above enlightenment only provides preliminary results, it has essential reference value for the planning and construction of DZs. First, the government should continue to promote the construction of DZs in various regions and give full play to the driving role of DZs in urbanization. The establishment of DZs should be subject to strict examination and approval management, and development zones with low developmental benefits and inconsistent standards should be addressed. In contrast, there should be a focus on maintaining the quantity and quality of DZs, strengthening their attraction to the population, and promoting their intensive land use and upgrading of industry. Second, the development mode of DZs should be changed and the construction level of their supporting facilities should be improved. DZs should not be regarded only as industrial agglomeration areas, but should also be built into new urban districts that complement the host city. By improving infrastructure construction and optimizing the living environment, DZs will attract the population and promote the development of urbanization. For example, public service facilities such as sports, medical care, and education will be enhanced in DZs and will improve residents' living environments. Third, the government should strengthen the spillover effect of DZs on the urbanization of surrounding areas. They should view DZs as a link between the city center and the suburbs. Moreover, the cooperation of transportation infrastructure between a DZ and its surrounding areas should be strengthened. By constructing a regionally integrated transportation network, the radiation effect of a DZ on its surrounding areas will be

enhanced. It is, therefore, of great significance to drive the urbanization development of the surrounding areas of a DZ.

This study provides empirical findings that deepen the understanding of the link between the urbanization process and DZ policy. These findings provide a vital policy reference for regions in formulating urbanization strategies. Compared with the existing research, the design of this study presents the following innovations. First, this study uses the DID model to explore the impact of the DZ policy on urbanization. It conducts propensity score matching analysis and a parallel trend test on the research sample. The results of the study exclude the influence of exogenous factors and have strong credibility. In addition, this study investigates the urbanization process from the perspectives of the population, land, and economy, which can reflect the comprehensive urbanization level. We believe this has important implications for identifying the sustainability of urbanization. Since there is no authoritative definition of sustainable urbanization, although this study attempts to apply multiple urbanization indicators, it still cannot fully reflect sustainable urbanization. In the future, more in-depth research on sustainable urbanization should be carried out based on sustainable development. Another limitation of this study was that it only focused on the correlation between urbanization and DZ policy, without conducting an in-depth discussion of the mechanisms. Based on this study, it might prove fruitful to further investigate the impact mechanism of DZs on urbanization from the perspectives of the environment, quality of life, social relations, services, and equity.

## 6. Conclusions

China has entered the "new normal" stage of its economic development and sustainable urbanization is essential to support China's economic transformation. Urbanization promoted by DZs not only leads to population agglomeration in cities, but also changes cities' land use and industrial structure. This study takes national DZs as the research object and empirically analyzes the impact of DZs on urbanization. Specifically, we collect the DZ and urbanization data of 235 cities in China from 1990 to 2017 and construct a DID model. Urbanization is a complex and comprehensive process. PU can reflect population agglomeration. LU can reflect the expansion of urban construction land. EU can reflect the contribution of secondary and tertiary industries to economic development. Urbanization is highly sustainable with the support of the population, infrastructure, and industries. Therefore, this study examines the impact of DZs on urbanization from the perspectives of the population, land, and economy, which is of great significance for judging the sustainable development of urbanization. In addition, China's regional development is significantly unbalanced. The spillover effects of DZs on urbanization also differ among regions. However, less attention has been paid to this issue in existing studies. This study divides the sample into three regions, namely the eastern, central, and western regions, to identify the differences among policy effects in different regions of China.

We found that after conducting propensity score matching, a parallel trend test, and controlling a series of variables, the DZ policy had no significant impact on PU. The establishment of DZs across the country reduced the LU level in their host cities by about 22.6%, while the EU level increased by about 3.2%. The results show that the DZ policy benefited the intensive use of construction land and the upgrading of the industrial structure to a certain extent. The regional regression results show that the impact of DZs on urbanization presented significant heterogeneity. For the eastern region, the DZ policy had no significant impact on PU but negatively impacted LU and EU. The LU level of cities with DZs was reduced by 6%, and the EU level was decreased by 1.2%. The results of this study support that the advantage of the DZ policy for the urbanization of the eastern region lies in the strengthening of the intensive use of construction land. For the central and western regions with relatively weak foundational development, DZs played a vital role in attracting the population and upgrading industries. However, DZs also increased the area of construction land in central and western cities, which may be detrimental to the intensive use of urban construction land.

**Author Contributions:** Conceptualization, S.W.; methodology, S.G. and K.L. (Kaimeng Li); software, S.G.; validation, S.G.; formal analysis, S.G.; investigation, S.G.; resources, S.G.; data curation, S.G. and Y.L.; writing—original draft preparation, K.L. (Kaimeng Li) and S.G.; writing—review and editing, K.L. (Kaimeng Li), S.G., Y.L., K.L. (Ke Luo) and S.W.; visualization, S.G.; supervision, S.W.; project administration, S.W.; funding acquisition, S.W. All authors have read and agreed to the published version of the manuscript.

**Funding:** This study was funded by the Humanities and Social Science Foundation of the Ministry of Education of China (no. 21YJAZH087).

**Institutional Review Board Statement:** Not applicable.

**Informed Consent Statement:** Not applicable.

**Data Availability Statement:** The data are contained within the article.

**Conflicts of Interest:** The authors declare no conflict of interest.

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
