# Peer review of "The Impact of Development Zones on China’s Urbanization from the Perspectives of the Population, Land, and the Economy"

_land, doi:10.3390/land11101726_

Round 1

Reviewer 1 Report

1. The topic feels incomplete. For example, How does land affect urbanization? If this paper is to investigate land effects on urbanization, it should be appropriate to use a spatial econometric model.

2. The limitations of the study should also be elaborated. In particular, the meaning of policies should be analyzed in depth.

3. The literature review is chaotic, and it is difficult for readers to understand its context. Many sentences have obvious non-native English expressions.

4. References cannot be displayed in a stacked fashion. For example. “Likewise, most studies on the labor market of DZs in the United 56 States do not yield positive conclusions [18-20]”.

5. Introduction: The objective of the paper presented need more clarifications to suit reader to understand the main idea of the paper. The research purpose and contributions of the article are unclear.

6. The discussion of empirical results is too brief. Why does this article have no policy recommendations?

7. There are too many basic grammar issues, it is better to spend more time on polishing the writings.

Author Response

5-September-2022

Dear Editor,

We are very grateful for having a chance to improve our manuscript “The impact of development zones on China's urbanization: Perspectives from population, land, and the economy” (Land-1889747). We also appreciate the editors and reviewers who reviewed our research and paid so much patience to our manuscript; the detailed comments and suggestions are very significant and helpful for the authors to improve the research.

Based on the comments and suggestions, we have made careful modifications to the manuscript. The main corrections in the paper and the responds to the reviewers’ comments are appended below. The detailed information can also be seen in our revised manuscript. Revised portions are marked in red color in the revised paper.

Although the authors have carefully improved the paper in accordance with the comments and suggestions, there may still exist some problems and errors in our revised manuscript. We invite the editors and referees to propose more criticisms and suggestions. We also hope the new manuscript will meet Land’s standard with approval.

Best regards.

Yours sincerely,

The Authors.

Response to Reviewers

Land-1889747

The impact of development zones on China's urbanization: Perspectives from population, land, and the economy

  1. Response to Reviewer #1:

(1) The topic feels incomplete. For example, how does land affect urbanization? If this paper is to investigate land effects on urbanization, it should be appropriate to use a spatial econometric model.

Response: Thank you very much for your reminder. The purpose of this paper is to explore whether the establishment of DZs affects urbanization, whether this effect is different among regions. Previous studies have not made an in-depth analysis of this topic. The results of this paper confirm that the establishment of DZs has a significant impact on urbanization. Based on this, we believe that it is vital and appropriate to continue to explore how DZs affect urbanization, which is also our future research direction. According to your reminder, we have added future research orientation in the discussion section. The additions are as follows.

In addition, another limitation of this paper was that it only focused on the correlation between urbanization and DZ policy without an in-depth discussion on the mechanism. Based on this study, it might prove fruitful to investigate further the impact mechanism of DZs on urbanization from the perspectives of environment, quality of life, social relations, services, and equity.

(2) The limitations of the study should also be elaborated. In particular, the meaning of policies should be analyzed in depth.

Response: Thank you very much for your reminder. According to your reminder, we have added the innovations and significance of this study in the conclusion section. In addition, we have also elaborated limitations of this paper and proposed our future research prospects. The relevant modifications are as follows.

This study has provided empirical findings which deepen the understanding of the link between the urbanization process and DZ policy. These findings provide a vital policy reference for regions in formulating urbanization strategies. Compared with the existing research, the design of this paper has the following innovations. First, this paper uses the DID model to explore the impact of the DZ policy on urbanization. It conducts propensity score matching analysis and parallel trend test on the research samples. The results of the study exclude the influence of exogenous factors and have strong credibility. In addition, this paper investigates the urbanization process from the perspectives of population, land, and economy, which can reflect the comprehensive urbanization level. We believe this has important implications for identifying the sustainability of urbanization. Since there is no authoritative definition of sustainable urbanization, although this paper tries to use multiple urbanization indicators, it still cannot fully reflect sustainable urbanization. In the future, more in-depth research on sustainable urbanization should be carried out based on sustainable development. In addition, another limitation of this paper was that it only focused on the correlation between urbanization and DZ policy without an in-depth discussion on the mechanism. Based on this study, it might prove fruitful to investigate further the impact mechanism of DZs on urbanization from the perspectives of environment, quality of life, social relations, services, and equity.

(3) The literature review is chaotic, and it is difficult for readers to understand its context. Many sentences have obvious non-native English expressions.

Response: Thank you very much for your reminder. We have reorganized the introduction part to be more logical. In the introduction, the first and second paragraphs respectively introduce the research status of world urbanization and China's urbanization and emphasized the importance of sustainable urbanization research. The third, fourth, and fifth paragraphs introduce the DZ model of urbanization and do a literature review on the policy effects of the DZ. The sixth paragraph clarifies the objective of this study, or the main innovation of this study. The related modifications of the introduction are as follows.

The emergence of cities is a sign of human maturity and civilization [1]. After two centuries of unprecedented rapid urbanization, more than half of the world's population lives in cities [2]. Rapid urbanization has a profound impact on global sustainability while increasing social productivity. The research on urbanization began at the beginning of the emergence of modern industrial cities and has been widely concerned by the academic community. Scholars believe that while urbanization promotes social modernization in various countries, it also brings a series of problems. For example, studies in London and New York found that state capital and planning play an essential role in urbanization [3]. In urban renewal, land values and rents have risen rapidly. However, ordinary citizens do not benefit from it, and the the wealth gap problem becomes more significant in urbanization [4]. The rapid development of urbanization has also deepened environmental issues and aroused academic attention to healthy and ecological cities [5]. Scholars have deeply analyzed urban air pollution [6], heat island effect [7], ecological pattern [8], land expansion [9], and other issues under the process of urbanization, and explored sustainable urbanization models.

Since 1990, China's urbanization has entered a stage of rapid development [10]. Urbanization drives not only large-scale population agglomeration to cities [11], but also promotes the urban spatial restructuring and industrial restructuring [12]. Sustainable urbanization is an important driving force and necessary condition for national modernization [13, 14]. For China, guiding sustainable urbanization not only determines the future of China's urbanization but also affects the prospects of world urbanization [15]. Therefore, it is crucial to explore the sustainable urbanization model. The academic community has not formed an authoritative definition of sustainable urbanization. With in-depth research on the process of urbanization in different disciplines, the concept of urbanization has also been extended [16]. Urbanization has gradually become a comprehensive process involving population migration, industrial structure adjustment, and land-use patterns transformation [17]. It is of great significance to identify the sustainable urbanization model from population, industry, and land perspectives.

Some scholars have summarized China's urbanization development into seven "promoting models". Among these, the "development zone (DZ) model" is one of the most representative [18]. The " DZ model" is a typical example of government-led urbanization. The government establishes DZ in regions with advantageous locations and provides a series of preferential policies [19]. China's first batch of DZs was all set up in eastern port cities with advantageous locations. To support the development of the DZs, the State gives preferential policies such as tax reduction and financial support. The DZ can accept the transfer of international capital and industries by its policy superiority and favorable investment environment [20]. Since establishing the first national DZ in 1984, DZs have become an important strategic initiative to promote local economic growth [21]. As of 2018, China has established a total of 552 national DZs. The location of DZs has also shifted from the eastern coastal regions to the relatively backward central and western regions. By establishing DZs, cities can complete industrial and population agglomeration [22], and achieve transformation of land structure and industrial structure [23]. Due to the superiority of the "DZ model", it is necessary to estimate the impact of DZs on urbanization.

Existing studies are less concerned with the impact of DZs on urbanization. The related research on DZs and urban economic development, land use, and labor markets provide a new perspective. British scholars found that establishing industrial space policies similar to DZs did not have an apparent driving effect on regional economic growth [24]. In contrast, the employment costs within the DZs were remarkably increased [25]. In addition, studies on DZs in the United States have found that although enterprises in DZs can enjoy tax deductions, employees' wages are lower [26]. Besides economic growth, the driving effect of DZs construction on regional employment has also attracted the attention of scholars [27]. Likewise, most studies on the labor market of DZs in the United States do not yield positive conclusions [28, 29]. However, research on the DZ policy in European countries has found that the construction of DZs can attract large numbers of the employed population [30]. At the same time, the regional high-tech industries have also been significantly developed [31].

For China, many studies have confirmed the positive effects of the DZ policy and promoted urban development through the following aspects. First, the establishment of the DZ is accompanied by large-scale infrastructure construction, which improves the spatial appearance of the city [32]. Second, the DZ attracts foreign investment into the city through preferential policies, forming industrial agglomerations and becoming the growth pole of the city [33]. Third, enterprises settled in the DZ increased not only urban tax revenue but also attracted labor accumulation [34]. Finally, DZs can increase urban productivity [35] and land-use efficiency [36] through their scale effect. Nevertheless, the development of China's DZs has not always been smooth. On the one hand, due to the excessive proportion of foreign investment in the DZs, they are affected by the international economic cycle. On the other hand, the abuse of preferential policies also leads to the spillover effects of DZs is no longer obvious [37]. Even between 1990 and 2005, due to the blind establishment of DZs by local governments, the phenomenon of "zone fever" frequently occurred in China [38]. The positive effect of the DZ is no longer significant. [39].

(4) References cannot be displayed in a stacked fashion. For example. “Likewise, most studies on the labor market of DZs in the United 56 States do not yield positive conclusions [18-20]”.

Response: Thank you very much for your suggestion. According to your suggestion, we have separated the quotes references. The relevant modifications are as follows.

Since 1990, China's urbanization has entered a stage of rapid development [10]. Urbanization drives not only large-scale population agglomeration to cities [11], but also promotes urban spatial restructuring and industrial restructuring [12].

With the in-depth research on the process of urbanization in different disciplines, the concept of urbanization has also been extended [16]. Urbanization has gradually become a comprehensive process involving population migration, adjustment of industrial structure, and transformation of land-use patterns [17].

The government establishes DZ in regions with advantageous locations and provides a series of preferential policies [19]. The DZ can accept the transfer of international capital and industries by virtue of its policy superiority and favorable investment environment [20].

Through the establishment of DZs, cities can complete industrial and population agglomeration [22], and achieve transformation of land structure and industrial structure [23].

British scholars found that establishing industrial space policies similar to DZs did not have an obvious driving effect on regional economic growth [24]. In contrast, the employment costs within the DZs were greatly increased [25].

However, research on the DZ policy in European countries has found that the construction of DZs can attract large numbers of the employed population [30], and significantly promote the development of regional high-tech industries [31].

Finally, DZs can increase urban productivity [35] and land-use efficiency [36] through their scale effect.

Even between 1990 and 2005, due to the blind establishment of DZs by local governments, the phenomenon of "zone fever" frequently occurred in China [38], which greatly reduced the positive impact of DZs [39].

(5) Introduction: The objective of the paper presented need more clarifications to suit reader to understand the main idea of the paper. The research purpose and contributions of the article are unclear.

Response: Thank you very much for your suggestion. The purpose of this paper is to explore China's sustainable urbanization model, which determines the future development prospects of China and even the world. Through analysis, we find that DZs play a significant role in the urbanization process, especially sustainable urbanization. Due to the superiority of the "DZ model", it is necessary to estimate the impact of DZs on urbanization. In addition, it is of great significance to identify the sustainable urbanization model from the perspectives of population, economy, and land. Therefore, this paper uses the DID model to quantify the role of DZs in China's urbanization from perspectives of population, land, and economy. According to your suggestion, we have summarized the shortcomings of existing research in the introduction. The research purpose and contributions of the article have been put forward. The relevant modifications are as follows.

To sum up, existing studies do not exclude the external environment's influence when analyzing DZ's policy effects of DZs. The policy effects of DZs on urbanization have not been thoroughly examined. Second, existing studies mostly use a single indicator to analyze the urbanization process, which cannot reflect the level of comprehensive urbanization and sustainable urbanization. Third, current studies have not considered spatial differences in the policy effects of DZs. Therefore, this paper uses the DID model to quantify the role of DZs in China's urbanization. We conduct a series of robustness analyses to exclude the influence of external circumstances on the results. Considering that urbanization is a comprehensive process, we evaluate the spillover effects of DZs on urbanization from the perspectives of population, land, and the economy. In addition, this paper identifies the heterogeneous influence of DZs on urbanization to serve targeted urbanization development policies.

(6) The discussion of empirical results is too brief. Why does this article have no policy recommendations?

Response: Thank you very much for your suggestion. According to your suggestion, we have reorganized and separated the conclusion and discussion sections of this article. We have also expanded the discussion section and put forward three policy suggestions for the planning and construction of the DZ. First, continue to promote the construction of DZs in various regions, and give full play to the driving role of DZs in urbanization. Second, change the development mode of the DZ and improve the construction level of its supporting facilities. Third, strengthen the spillover effect of DZs on the urbanization of surrounding areas. The relevant modifications are as follows.

  1. Discussion

China has entered the "new normal" stage of economic development. Sustainable urbanization is essential to support China's economic transformation. Urbanization promoted by DZs not only leads to population agglomeration in cities but also changes cities' land use and industrial structure. This paper takes national DZs as the research object and empirically analyzes the impact of DZs on urbanization. Specifically, we collected the DZs and urbanization data of 235 cities in China from 1990 to 2017 and constructed a DID model to carry out the study. Urbanization is a complex and comprehensive process. PU can reflect population agglomeration. LU can reflect the expansion of urban construction land. EU can reflect the contribution of secondary and tertiary industries to economic development. Urbanization is highly sustainable with the support of population, infrastructure, and industries. Therefore, this paper examines the impact of DZs on urbanization from the perspectives of population, land, and economy, which is of great significance for judging the sustainable development of urbanization. In addition, China's regional development is significantly unbalanced. The spillover effects of DZs on urbanization are also different among regions. However, less attention has been paid to this issue in existing studies. This paper divides the sample into three regions, namely the eastern, central, and western regions, to identify the differences among policy effects in different regions of China.

The results show that the DZ policy benefited the intensive use of construction land and the upgrading of the industrial structure to a certain extent. The regional regression results show that the impact of DZs on urbanization presented significant heterogeneity. The results of this study support that the advantage of the DZ policy for the urbaniza-tion of the eastern region was to strengthen the intensive use of construction land. For the central and western regions with relatively weak foundational development, DZs had played a vital role in attracting population and upgrading industries.  However, DZs also increased the area of construction land in central and western cities, which may be detrimental to the intensive use of urban construction land.

Although the above enlightenment is only a preliminary conclusion, it has essential reference value for the planning and construction of the DZ. First, continue to promote the construction of DZs in various regions and give full play to the driving role of DZs in urbanization. The establishment of DZs shall be subject to strict examination and approval management, and the development zones with low developmental benefits and inconsistent standards shall be addressed. In contrast, maintaining the quantity and quality of the DZ, strengthening its attraction to the population, and promoting its intensive land use and up-grading of industry. Second, change the development mode of the DZ and improve the construction level of its supporting facilities. The DZ is regarded not only as an industrial agglomeration area but is built into a new urban district that complements the host city. By improving infrastructure construction and optimizing the living environment, the DZ will attract population gathering and promote the development of urbanization. For example, enhance public service facilities such as sports, medical care, and education in the DZ and improve its living functions. Third, strengthen the spillover effect of DZs on the urbanization of surrounding areas. View the DZ as a link between the city center and the suburbs. Strengthen the cooperation of transportation infrastructure between the DZ and its surrounding areas. By constructing a regionally integrated transportation network, the radiation effect of the DZ on the surrounding areas will be enhanced. It is of great significance to drive the urbanization development of the surrounding areas of the DZ.

  1. Conclusion

Using the DID model, this paper focuses on analyzing the spillover effect of the DZ policy on urbanization. We found that after conducting propensity score matching, a parallel trend test, and controlling a series of variables, the DZ policy had no significant impact on PU. The establishment of DZs across the country had reduced the LU level in their host cities by about 22.6%, while the EU level had increased by about 3.2%. For the eastern region, the DZ policies had no significant impact on PU but negatively impacted LU and EU. The LU level of the cities with DZs was reduced by 6%, and the EU level was decreased by 1.2%. For the central and western regions with relatively weak foundational development, DZs played a vital role in attracting the population and upgrading industries. However, DZs also increased the area of construction land in central and western cities, which may be detrimental to the intensive use of urban construction land.

This study has provided empirical findings which deepen the understanding of the link between the urbanization process and DZ policy. These findings provide a vital policy reference for regions in formulating urbanization strategies. Compared with the existing research, the design of this paper has the following innovations. First, this paper uses the DID model to explore the impact of the DZ policy on urbanization. It conducts propensity score matching analysis and parallel trend test on the research samples. The results of the study exclude the influence of exogenous factors and have strong credibility. In addition, this paper investigates the urbanization process from the perspectives of population, land, and economy, which can reflect the comprehensive urbanization level. We believe this has important implications for identifying the sustainability of urbanization. Since there is no authoritative definition of sustainable urbanization, although this paper tries to use multiple urbanization indicators, it still cannot fully reflect sustainable urbanization. In the future, more in-depth research on sustainable urbanization should be carried out based on sustainable development. In addition, another limitation of this paper was that it only focused on the correlation between urbanization and DZ policy without an in-depth discussion on the mechanism. Based on this study, it might prove fruitful to investigate further the impact mechanism of DZs on urbanization from the perspectives of environment, quality of life, social relations, services, and equity.

(7) There are too many basic grammar issues, it is better to spend more time on polishing the writings.

Response: Thank you very much for your reminder. We have polished the writings and checked the text for possible grammar issues. We have used English editing service to polish the manuscript and provided a polished manuscript as proof of editing.

  1. Response to Reviewer #2:

The purpose of the authors is to take the panel data of 235 cities in China from 1990 to 2017 to evaluate the policy effects of setting up development zones on urbanization from the perspectives of the population, land, and the economy. The topic is significant. Submitted manuscript is written in high quality English language. Research questions are clearly formulated, so it is easy to understand the aim of this research. Methodology used in this research is appropriate. The manuscript is full of valuable Figures and Tables which clearly demonstrate the strength of the paper. I suggest to improve the text, update the References, write a separate Discussion section and improving the Conclusion and accepting after this.

(1) The formulas are not correctly displayed and need to be rewritten. The formulas on line 99-101 need to be rewritten because they are displayed badly in the text, more explanation is needed.

Response: Thank you very much for your suggestion. This paper uses the DID model to evaluate the impact of DZ on urbanization. There are three dependent variables in this article. Therefore, we use three formulas to express the DID model. These formulas are the same except for dependent variables. This article does not explain the elements of each formula separately, which can cause formulas to display badly in the text. According to your suggestion, we combine the formulas of line 99-101 and line 213-215 to facilitate readers to better understanding. The modified formulas and explanations are as follows.

l                      (1)

where  is the dependent variable, which represents the population urbanization (PU), land urbanization (LU), and economic urbanization (EU) levels of city  at time , respectively;  is the independent variable, which indicates whether city  has set up a DZ at time ; represents the control variables such as per capita GDP (PcGDP), average wage (AS), budgetary revenue (BR), fixed asset in-vestment (FAI), foreign direct investment (FDI), political status (PS), nearshore distance (ND), and latitude (LA); , , and  are regression coefficients, where  is the focus of this paper and represents the impact of DZ policy on urbanization;  and  are the fixed effects on the time and individual, respectively; and  is a random disturbance term. In order to eliminate the possible heteroscedasticity in the model, the dependent variables and the control variable are processed logarithmically in this paper. In addition, this paper divides the sample into three regions, namely the eastern, central, and western regions, to identify the differences among policy effects in different regions of China.

     (2)

where  is the dependent variable, which represents the population urbanization (PU), land urbanization (LU), and economic urbanization (EU) levels of city  at time , respectively; where  is the dummy variable for establishing the DZ;  indicates that time  is 5 years before the establishment of the DZ in city , it is set as 1; otherwise, it is set as 0;  indicates that if time  is 5 years after the establishment of the DZ in city , it is set as 1; otherwise, it is set as 0; and other dummy variables are defined similarly;  and  are the fixed effects on time and the individual, respectively; and  is a random disturbance term.

(2) I suggest slightly bigger description of the Figure 1 and Figure 2, it is difficult to read it.

Response: Thank you very much for your suggestion. According to your suggestion, we have added descriptions of Figure 1 and Figure 2. The additions are as follows.

Figure 1 displays the spatial pattern of the PU, LU, and EU levels for the treatment and control groups in 1990 and 2017. From Figure 1, the PU, LU, and EU levels varied significantly among cities. The difference in LU level between cities was the most significant, followed by PU. And the EU level was relatively balanced among cities. These reflected the complexity of the urbanization process. Therefore, it was necessary to measure the impact of DZs on urbanization from different perspectives. On the whole, the urbanization level in the eastern coastal regions was observably higher than that in the central and western regions. The Beijing-Tianjin-Hebei Urban Agglomerations, Yangtze River Delta Urban Agglomerations, and Pearl River Delta Urban Agglomerations were the high-value areas of urbanization level, and the increase was the most obvious during the study period. The regional difference in urbanization level proved that it was reasonable to divide the sample into three regions. In addition, Figure 1 further confirmed that the urbanization level of the treatment group cities was higher than that of the control group.

Figure 1. Urbanization spatial pattern of treatment group and control group in China

As can be seen from Figure 2, the establishment of DZs exerted an impact on urbanization. And the impact of DZs on urbanization increases significantly over time. The policy effect on urbanization in years without DZs fluctuated around the zero value. This indicated no unparallel trend in the sample before the city set up the DZ. However, after the establishment of DZs, the policy effect of PU, LU, and EU significantly deviated from a zero value. It showed that the establishment of DZs exerted an impact on urbanization progress. And the treatment and control groups in this paper also passed the parallel trend test. The establishment of DZs in cities significantly affected the inherent processes of PU, LU, and EU. Specifically, the DZ policy exerted an incredibly positive effect on PU and EU, while it negatively affected LU. In addition, Figure 2 showed that the policy effect of DZs on urbanization increases significantly over time. Because of this, we have reason to believe that establishing DZs can significantly change the developmental trend of the urban population, land use, and industrial industry, and these impacts will become more pronounced over time.

Figure 2. Parallel trend test plots

(3) On some places the quality of the English need to be improved.

Response: Thank you very much for your reminder. We have polished the writings and checked the text for possible grammar issues. We have used English editing service to polish the manuscript and provided a polished manuscript as proof of editing.

(4) When you talk about the development process of urbanization that invlolves other fields on line 122 and below I strongly suggest to say a few words about global health and ecological security and to cite these papers in the field:

  • Todorov, V.; Dimov, I. Innovative Digital Stochastic Methods for Multidimensional Sensitivity Analysis in Air Pollution Modelling. Mathematics 2022, 10, 2146. https://doi.org/10.3390/math10122146
  • Piracha, A.; Chaudhary, M.T. Urban Air Pollution, Urban Heat Island and Human Health: A Review of the Literature. Sustainability 2022, 14, 9234. https://doi.org/10.3390/su14159234
  • Liu, H.; Li, Q.; Yu, D.; Gu, Y. Air Quality Index and Air Pollutant Concentration Prediction Based on Machine Learning Algorithms. Appl. Sci. 2019, 9, 4069. https://doi.org/10.3390/app9194069

Response: Thank you very much for your suggestion. According to your suggestion, we have discussed global health and ecological security issues in the process of urbanization in the introduction. In addition, we have added the discussion on urban capital and land sprawl under the process of urbanization, citing international literature in these fields. The additions are as follows.

The emergence of cities is a sign of human maturity and civilization [1]. After two centuries of unprecedented rapid urbanization, more than half of the world's population lives in cities [2]. Rapid urbanization has a profound impact on global sustainability while increasing social productivity. The research on urbanization began at the beginning of the emergence of modern industrial cities and has been widely concerned by the academic community. Scholars believe that while urbanization promotes social modernization in various countries, it also brings a series of problems. For example, studies in London and New York found that state capital and planning play an essential role in urbanization [3]. In urban renewal, land values and rents have risen rapidly. However, ordinary citizens do not benefit from it, and the the wealth gap problem becomes more significant in urbanization [4]. The rapid development of urbanization has also deepened environmental issues and aroused academic attention to healthy and ecological cities [5]. Scholars have deeply analyzed urban air pollution [6], heat island effect [7], ecological pattern [8], land expansion [9], and other issues under the process of urbanization, and explored sustainable urbanization models.

References:

  1. Turok, I.; Mykhnenko, V., The trajectories of European cities, 1960–2005. Cities 2007, 24, (3), 165-182.
  2. Mulligan, G. F., Reprint of: Revisiting the urbanization curve. Cities 2013, 32, S58-S67.
  3. Stein, S., Capital city: Gentrification and the real estate state. Verso Books: 2019.
  4. Atkinson, R., Alpha city: How London was captured by the super-rich. Verso Books: 2021.
  5. Mueller, N.; Rojas-Rueda, D.; Khreis, H.; Cirach, M.; Andrés, D.; Ballester, J.; Bartoll, X.; Daher, C.; Deluca, A.; Echave, C., Changing the urban design of cities for health: The superblock model. Environment international 2020, 134, 105132.
  6. Todorov, V.; Dimov, I., Innovative digital stochastic methods for multidimensional sensitivity analysis in air pollution modelling. Mathematics 2022, 10, (12), 2146.
  7. Piracha, A.; Chaudhary, M. T., Urban air pollution, urban heat island and human health: A review of the literature. Sustainability 2022, 14, (15), 9234.
  8. Liu, H.; Li, Q.; Yu, D.; Gu, Y., Air quality index and air pollutant concentration prediction based on machine learning algorithms. Applied Sciences 2019, 9, (19), 4069.
  9. Jarah, S. H. A.; Zhou, B.; Abdullah, R. J.; Lu, Y.; Yu, W., Urbanization and urban sprawl issues in city structure: A case of the Sulaymaniah Iraqi Kurdistan Region. Sustainability 2019, 11, (2), 485.

(5) The weak point of this article is precisely the way it is written, which jeopardizes its following by readers, especially in a text of this complexity. A careful revision of the text is suggested, in order to use shorter sentences (some of the current ones should be divided), use commas and periods judiciously to clearly separate the ideas presented, and pay some attention to grammatical details.

Response: Thank you very much for your suggestion. According to your suggestion, we have revised the text carefully. We have broken down some long sentences into shorter sentences. In addition, we have also checked the text for possible grammar issues. We have used English editing service to polish the manuscript and provided a polished manuscript as proof of editing.

(6) I strongly suggest two separate sections Discussion and Conclusion, and more focus on Discussion section, Section 4 in the current form is unacceptable!

Response: Thank you very much for your suggestion. According to your suggestion, we have reorganized and separated this article's conclusion and discussion section. In the discussion section, we focus on the purpose and significance of this study and put forward three policy suggestions for the planning and construction of the DZ. First, continue to promote the construction of DZs in various regions and give full play to the driving role of DZs in urbanization. Second, change the DZ's development mode and improve its supporting facilities' construction level. Third, strengthen the spillover effect of DZs on the urbanization of surrounding areas. The relevant modifications are as follows.

  1. Discussion

China has entered the "new normal" stage of economic development. Sustainable urbanization is essential to support China's economic transformation. Urbanization promoted by DZs not only leads to population agglomeration in cities but also changes cities' land use and industrial structure. This paper takes national DZs as the research object and empirically analyzes the impact of DZs on urbanization. Specifically, we collected the DZs and urbanization data of 235 cities in China from 1990 to 2017 and constructed a DID model to carry out the study. Urbanization is a complex and comprehensive process. PU can reflect population agglomeration. LU can reflect the expansion of urban construction land. EU can reflect the contribution of secondary and tertiary industries to economic development. Urbanization is highly sustainable with the support of population, infrastructure, and industries. Therefore, this paper examines the impact of DZs on urbanization from the perspectives of population, land, and economy, which is of great significance for judging the sustainable development of urbanization. In addition, China's regional development is significantly unbalanced. The spillover effects of DZs on urbanization are also different among regions. However, less attention has been paid to this issue in existing studies. This paper divides the sample into three regions, namely the eastern, central, and western regions, to identify the differences among policy effects in different regions of China.

The results show that the DZ policy benefited the intensive use of construction land and the upgrading of the industrial structure to a certain extent. The regional regression results show that the impact of DZs on urbanization presented significant heterogeneity. The results of this study support that the advantage of the DZ policy for the urbaniza-tion of the eastern region was to strengthen the intensive use of construction land. For the central and western regions with relatively weak foundational development, DZs had played a vital role in attracting population and upgrading industries.  However, DZs also increased the area of construction land in central and western cities, which may be detrimental to the intensive use of urban construction land.

Although the above enlightenment is only a preliminary conclusion, it has essential reference value for the planning and construction of the DZ. First, continue to promote the construction of DZs in various regions and give full play to the driving role of DZs in urbanization. The establishment of DZs shall be subject to strict examination and approval management, and the development zones with low developmental benefits and inconsistent standards shall be addressed. In contrast, maintaining the quantity and quality of the DZ, strengthening its attraction to the population, and promoting its intensive land use and up-grading of industry. Second, change the development mode of the DZ and improve the construction level of its supporting facilities. The DZ is regarded not only as an industrial agglomeration area but is built into a new urban district that complements the host city. By improving infrastructure construction and optimizing the living environment, the DZ will attract population gathering and promote the development of urbanization. For example, enhance public service facilities such as sports, medical care, and education in the DZ and improve its living functions. Third, strengthen the spillover effect of DZs on the urbanization of surrounding areas. View the DZ as a link between the city center and the suburbs. Strengthen the cooperation of transportation infrastructure between the DZ and its surrounding areas. By constructing a regionally integrated transportation network, the radiation effect of the DZ on the surrounding areas will be enhanced. It is of great significance to drive the urbanization development of the surrounding areas of the DZ.

(7) You should add in the Conclusion about the novelties of the originality of the observed results mentioned already in the Discussion section. A few words about future work and to whom the survey will be important will be valuable.

Response: Thank you very much for your reminder. According to your reminder, we have added the innovations and significance of this study in the conclusion section. In addition, we have also elaborated limitations of this paper and proposed our future research prospects. The relevant modifications are as follows.

  1. Conclusion

Using the DID model, this paper focuses on analyzing the spillover effect of the DZ policy on urbanization. We found that after conducting propensity score matching, a parallel trend test, and controlling a series of variables, the DZ policy had no significant impact on PU. The establishment of DZs across the country had reduced the LU level in their host cities by about 22.6%, while the EU level had increased by about 3.2%. For the eastern region, the DZ policies had no significant impact on PU but negatively impacted LU and EU. The LU level of the cities with DZs was reduced by 6%, and the EU level was decreased by 1.2%. For the central and western regions with relatively weak foundational development, DZs played a vital role in attracting the population and upgrading industries. However, DZs also increased the area of construction land in central and western cities, which may be detrimental to the intensive use of urban construction land.

This study has provided empirical findings which deepen the understanding of the link between the urbanization process and DZ policy. These findings provide a vital policy reference for regions in formulating urbanization strategies. Compared with the existing research, the design of this paper has the following innovations. First, this paper uses the DID model to explore the impact of the DZ policy on urbanization. It conducts propensity score matching analysis and parallel trend test on the research samples. The results of the study exclude the influence of exogenous factors and have strong credibility. In addition, this paper investigates the urbanization process from the perspectives of population, land, and economy, which can reflect the comprehensive urbanization level. We believe this has important implications for identifying the sustainability of urbanization. Since there is no authoritative definition of sustainable urbanization, although this paper tries to use multiple urbanization indicators, it still cannot fully reflect sustainable urbanization. In the future, more in-depth research on sustainable urbanization should be carried out based on sustainable development. In addition, another limitation of this paper was that it only focused on the correlation between urbanization and DZ policy without an in-depth discussion on the mechanism. Based on this study, it might prove fruitful to investigate further the impact mechanism of DZs on urbanization from the perspectives of environment, quality of life, social relations, services, and equity.

  1. Response to Reviewer #3:

The paper aims to assess the impact of establishing development zones on urbanization development using a static method known as difference-in-differences. It is a good paper and it is based on a solid statistical approach but in my vision it is lacking in reflection. I think it could appears more interesting for a researcher in economics than in urban planning. Furthermore, authors focus on quantifying impacts of DZ zones but a not considered question is: what about the quality of the impact? In any case, I think some improvements can be made.

(1) In row 11 authors wrote: “The sustainable development of urbanization is a necessary condition for China to realize modernization”. But the analysis made is not focused on sustainability. Sustainability is nowhere else mentioned in the paper, and there is no a definition of it.

Response: Thank you very much for your reminder. Although this paper does not focus on "sustainability", the topic and content of the study are all related to sustainable urbanization. On the one hand, the dependent variable of this paper is not only population urbanization (PU), but also land urbanization (LU) and economic urbanization (EU). LU can reflect the expansion of urban construction land. EU can reflect the contribution of secondary and tertiary industries to economic development. With the support of the population, infrastructure, and industries, urbanization is highly sustainable. Therefore, we believe that multiple perspectives can reflect the sustainability of urbanization development to a certain extent. On the other hand, the independent variable of this paper is the DZ policy. Previous studies have confirmed that urbanization development promoted by DZs has strong sustainability ("Li, Q.; Chen, Y.; Liu, J. On the development mode of Chinese urbanization. Social Sciences in China 2012, 7, 82-100 "). Because the DZ is the regional industrial agglomeration center and the growth pole of economic development, it has strong development vitality. The empirical results of this paper show that the DZ does not positively impact the PU, LU, and EU in all regions. Therefore, this paper puts forward policy suggestions to strengthen the role of DZ in promoting urbanization to realize "sustainable" urbanization. Due to the lack of definitions and standards of sustainable urbanization, the "sustainability" analysis of urbanization in this paper is still insufficient. According to your reminder, we take the analysis of sustainable urbanization as the research deficiency and future research prospect of this paper. The relevant modifications are as follows.

In addition, this paper investigates the urbanization process from the perspectives of population, land, and economy, which can reflect the comprehensive urbanization level. We believe this has important implications for identifying the sustainability of urbanization. Since there is no authoritative definition of sustainable urbanization, although this paper tries to use multiple urbanization indicators, it still cannot fully reflect sustainable urbanization. In the future, more in-depth research on sustainable urbanization should be carried out based on sustainable development.

(2) The DZ model is the core of the paper, so I suggest looking into the subject in more depth. More information would be appreciated (for example what purpose they serve, through which policies they were introduced and when, how the areas were selected, what criteria were used). (Introduction)

Response: Thank you very much for your suggestion. The "DZ model" is a typical example of government-led urbanization. And the purpose of DZ is to accept the transfer of international capital and industries. According to your suggestion, we have added the relevant expressions of the DZ model, including its establishment purpose, preferential policies, spatial layout, development objectives, development status, etc. The relevant modifications are as follows.

Some scholars have summarized China's urbanization development into seven "promoting models". Among these, the "development zone (DZ) model" is one of the most representative [18]. The " DZ model" is a typical example of government-led urbanization. The government establishes DZ in regions with advantageous locations and provides a series of preferential policies [19]. China's first batch of DZs was all set up in eastern port cities with advantageous locations. To support the development of the DZs, the State gives preferential policies such as tax reduction and financial support. The DZ can accept the transfer of international capital and industries by its policy superiority and favorable investment environment [20]. Since establishing the first national DZ in 1984, DZs have become an important strategic initiative to promote local economic growth [21]. As of 2018, China has established a total of 552 national DZs. The location of DZs has also shifted from the eastern coastal regions to the relatively backward central and western regions. By establishing DZs, cities can complete industrial and population agglomeration [22], and achieve transformation of land structure and industrial structure [23]. Due to the superiority of the "DZ model", it is necessary to estimate the impact of DZs on urbanization.

(3) In row 80 authors wrote: “Therefore, this paper uses the DID model to quantify the role of DZs in China's urbanization to fill the above research gaps”. I suggest making it explicitly clear what gaps are intended to be bridged by the analysis”.

Response: Thank you very much for your suggestion. According to your suggestion, we have summarized three shortcomings of existing research. First, existing studies do not exclude the influence of the external environment. Second, existing studies mostly use a single indicator to analyze the urbanization process, which cannot reflect the level of comprehensive urbanization and sustainable urbanization. Third, current studies have not considered spatial differences in the policy effects of DZs. Therefore, we have made a targeted research design to fill the above gaps. The relevant modifications are as follows.

To sum up, existing studies do not exclude the external environment's influence when analyzing DZ's policy effects of DZs. The policy effects of DZs on urbanization have not been thoroughly examined. Second, existing studies mostly use a single indicator to analyze the urbanization process, which cannot reflect the level of comprehensive urbanization and sustainable urbanization. Third, current studies have not considered spatial differences in the policy effects of DZs. Therefore, this paper uses the DID model to quantify the role of DZs in China's urbanization. We conduct a series of robustness analyses to exclude the influence of external circumstances on the results. Considering that urbanization is a comprehensive process, we evaluate the spillover effects of DZs on urbanization from the perspectives of population, land, and the economy. In addition, this paper identifies the heterogeneous influence of DZs on urbanization to serve targeted urbanization development policies.

(4) In row 89 authors wrote “using the DID method”. The acronym is introduced without giving a definition.

Response: Thank you very much for your reminder. The abstract has defined DID in row 15. According to your reminder, we have added the definition of DID in row 114 to make it easier for readers to understand.

(5) In row 91 authors wrote “from the perspectives of the population, land, and the economy”. It seems a too general sentence, I suggest clarifying which aspects of these elements (population, land, economy) authors intend to investigate.

Response: Thank you very much for your suggestion. The development process of urbanization involves the population, land, economy, and other fields. Therefore, this paper evaluates the spillover effects of DZ on urbanization from the perspectives of the population, land, and the economy. Specifically, when exploring the spillover effects of DZ policy on urbanization, the dependent variables comprehensively consider population urbanization (PU), land urbanization (LU), and economic urbanization (EU). The Variables and Data section has clarified these elements (population, land, economy). Include variable definitions and data sources. The relevant contents are as follows.

Urbanization is not only manifested in the transfer of the rural population to the urban population but is also accompanied by the adjustment of industrial structure and the spread of construction land. The development process of urbanization involves the population, land, economy, and other fields [45, 46]. When exploring the spillover effects of DZ policy, the dependent variables comprehensively consider population urbanization (PU), land urbanization (LU), and economic urbanization (EU). PU is the ratio of the urban population to the total population of a city. LU is the ratio of urban construction land area to the total area. EU is the proportion of the output value of the city's secondary and tertiary industries to its GDP. The population data comes from the "China Regional Statistical Yearbook 1990–2018" The urban construction land data comes from the Environmental Science Data Center of the Chinese Academy of Sciences, and the economic data comes from the "China City Statistical Yearbook 1990–2018".

We think it would be more appropriate to clarify these elements in the Variables and Data section. Therefore, we have deleted the statement in section 2.1 and clarified it in section 2.2. We hope our response can be accepted and welcome your further suggestions.

(6) Please clarify why the DID model was chosen and how it works from a conceptual point of view (for those who are not statisticians).

Response: Thank you very much for your suggestion. According to your suggestion, we have revised the clarification of the DID model. We have added reasons for choosing the DID model. And we also clarify the conceptual point of the DID model so that non-statisticians can better understand the method. The modifications are as follows.

This paper aims to test whether the development zone policy can promote the urbanization process. The difference in difference (DID) model is practical for evaluating policy effects. The model divides the research sample into treatment group (where the policy is implemented) and control group (where the policy is not implemented). Unobservable factors are eliminated by differentiating before and after policy implementation and between the treatment group and the control group, and the policy net effect can be identified. Scholars usually use the DID model to conduct policy evaluation because this model can better avoid the endogenous effects of policies [40-42]. This paper divides the research sample into a treatment group and a control group according to whether the city has already established a DZ. This allows the DID model to be constructed based on the double fixed effects of time and the individual. Because the time of establishing DZs in various cities is not consistent, this article chooses the progressive DID model to test the impact of DZ on urbanization [43, 44]. The expression of the model is as follows:

l                      (1)

where  is the dependent variable, which represents the population urbanization (PU), land urbanization (LU), and economic urbanization (EU) levels of city  at time , respectively;  is the independent variable, which indicates whether city  has set up a DZ at time ;  represents the control variables such as per capita GDP (PcGDP), average wage (AS), budgetary revenue (BR), fixed asset in-vestment (FAI), foreign direct investment (FDI), political status (PS), nearshore distance (ND), and latitude (LA); , , and  are regression coefficients;  and  are the fixed effects on the time and individual, respectively; and  is a random disturbance term. In order to eliminate the possible heteroscedasticity in the model, the dependent variables and the control variable are processed logarithmically in this paper. In the model regression, the coefficient  is the focus of this paper. The coefficient reflects the impact of the DZ policy on the urbanization process after double difference. If  is significantly positive, it means that the DZ policy can promote the regional urbanization process.

(7) Why are the 'propensity score matching' and 'kernel matching method' not mentioned in this paragraph on the method?

Response: Thank you very much for your reminder. The purpose of propensity score matching is to test the robustness of DID model. The kernel matching method is one of many matching methods for propensity score matching. Similarly, parallel trend analysis is also a method to test the robustness of DID model. These methods are not the core methods of this paper. Therefore, we think it appropriate to express these methods as a robustness test in the result section. Previous studies often used this presentation. Accordance to your reminder, we have added expressions related to propensity score matching analysis and parallel trend test in the methodology section. The additions are as follows.

The DID model requires that the division of the treatment group and the control group is random. And the treatment group and the control group should also meet the parallel trend assumption before implementing the policy. Therefore, this paper conducts propensity score matching analysis and parallel trend test on the sample to ensure the accuracy of the estimation results. In addition, this paper divides the sample into the eastern, central, and western regions, to identify the differences among policy effects in different regions of China.

(8) Furthermore, measuring social development solely by a purely economic parameter/s does not seem sufficient.

Response: Thank you very much for your reminder. The development of urbanization is not only affected by the DZ policy but also by other factors. Therefore, this paper introduces some control variables into the DID model. These variables are important factors affecting the urbanization process rather than measuring social development. Accordance to your reminder, selecting control variables solely by purely economic parameters does not seem sufficient. Therefore, in the selection of control variables, in addition to economic parameters, we also selected non-economic variables such as political status, nearshore distance, and latitude. We believe that considering both economic and non-economic parameters will make the model more accurate. We hope our response can be accepted and welcome your further suggestions.

(9) There is no critical mention of the fact that the effects of urbanisation should also include issues such as the environment, quality of life, social relations, services, land consumption, climate effects, equity, etc. in their assessment.

Response: Thank you very much for your suggestion. We are entirely in agreement with you that the impact of urbanization involves many aspects. The issues you mentioned, such as the environment, quality of life, social relations, services, land consumption, and climate impact, are also hotspots in urbanization research. According to your suggestion, we believe that the impact mechanism of DZ on urbanization can be discussed from the perspectives of environment, quality of life, social relations, and services in the future. This perspective is interesting and important. Therefore, we have taken it as the future research prospect of this paper.

Another limitation of this paper was that it only focused on the correlation between urbanization and DZ policy without an in-depth discussion on the mechanism. Based on this study, it might prove fruitful to investigate further the impact mechanism of DZs on urbanization from the perspectives of environment, quality of life, social relations, services, and equity.

(10) In row 346 authors wrote: “High-quality urbanization is an important support mechanism for China's sustainable development”. However I think that the parameters used are not adequate to define quality. So what do you mean by “high quality urbanization”?

Response: Thank you very much for your suggestion. We believe that the urbanization process promoted by the DZ is characterized by high quality and sustainability. Because the DZ is the regional industrial agglomeration center and the growth pole of economic development, it has strong development vitality. Urbanization promoted by DZs not only leads to population agglomeration in cities but also changes the land use and industrial structure of cities, which can significantly improve the development quality of urbanization. According to your suggestion, we have decided to delete the expression of high-quality urbanization and replace it with sustainable urbanization. The reasons for using sustainable urbanization have been explained in our response to question 1. We hope the response can be accepted and welcome your further suggestions.

  1. Response to Reviewer #4:

This text is a good piece of paper and it gives more insights policy effects of setting up development zones on urbanization from the perspectives of the population, land, and the economy because it evaluates the policies effects. Nevertheless, the paper is a China-based case study that needs more reflection for the international readership. I highly suggest to add a paragraph or a new text in the introduction with the international literature review in the following aspects:

1) the distortions provoked by capitalist urban regeneration processes by taking into references some examples of global cities:

- 2020. Alpha City: How London Was Captured by the Super-Rich. London: Verso

- 2019. Capital City. Gentrification and the real estate state. London-New York: Verso

- 2019. Regenerating Bilbao: From "productive industries" to "productive services". Territorio, 89

2) healthy cities to contrast speculative urbanization:

- 2020. Changing the urban design of cities for health: The superblock model. Environment International, 134: 105132

3) Urbanization and urban sprawl:

https://www.mdpi.com/2071-1050/11/2/485

4) the indirect effects of urbanization in the redevelopment of historic neighborhoods

- 2019. Historic Cities: Issues in Urban Conservation, The Getty Conservation Institute, Los Angeles.

The analysis outstands but needs a more critical literature review. By doing so, the paper will be suitable for publishing and for international readership. Also, write the lessons learned of this analysis in the conclusion. The conclusion is not sufficient for the analysis done by Authors. This is why I require a new (and updated) version of the paper.

Response: Thank you very much for your suggestion. According to your suggestion, we have added the discussion on the urban regeneration process, health of cities and ecological security, urban capital, and land sprawl under the process of urbanization, citing international literature in these fields. The additions are as follows.

The emergence of cities is a sign of human maturity and civilization [1]. After two centuries of unprecedented rapid urbanization, more than half of the world's population lives in cities [2]. Rapid urbanization has a profound impact on global sustainability while increasing social productivity. The research on urbanization began at the beginning of the emergence of modern industrial cities and has been widely concerned by the academic community. Scholars believe that while urbanization promotes social modernization in various countries, it also brings a series of problems. For example, studies in London and New York found that state capital and planning play an essential role in urbanization [3]. In urban renewal, land values and rents have risen rapidly. However, ordinary citizens do not benefit from it, and the the wealth gap problem becomes more significant in urbanization [4]. The rapid development of urbanization has also deepened environmental issues and aroused academic attention to healthy and ecological cities [5]. Scholars have deeply analyzed urban air pollution [6], heat island effect [7], ecological pattern [8], land expansion [9], and other issues under the process of urbanization, and explored sustainable urbanization models.

References:

  1. Turok, I.; Mykhnenko, V., The trajectories of European cities, 1960–2005. Cities 2007, 24, (3), 165-182.
  2. Mulligan, G. F., Reprint of: Revisiting the urbanization curve. Cities 2013, 32, S58-S67.
  3. Stein, S., Capital city: Gentrification and the real estate state. Verso Books: 2019.
  4. Atkinson, R., Alpha city: How London was captured by the super-rich. Verso Books: 2021.
  5. Mueller, N.; Rojas-Rueda, D.; Khreis, H.; Cirach, M.; Andrés, D.; Ballester, J.; Bartoll, X.; Daher, C.; Deluca, A.; Echave, C., Changing the urban design of cities for health: The superblock model. Environment international 2020, 134, 105132.
  6. Todorov, V.; Dimov, I., Innovative digital stochastic methods for multidimensional sensitivity analysis in air pollution modelling. Mathematics 2022, 10, (12), 2146.
  7. Piracha, A.; Chaudhary, M. T., Urban air pollution, urban heat island and human health: A review of the literature. Sustainability 2022, 14, (15), 9234.
  8. Liu, H.; Li, Q.; Yu, D.; Gu, Y., Air quality index and air pollutant concentration prediction based on machine learning algorithms. Applied Sciences 2019, 9, (19), 4069.
  9. Jarah, S. H. A.; Zhou, B.; Abdullah, R. J.; Lu, Y.; Yu, W., Urbanization and urban sprawl issues in city structure: A case of the Sulaymaniah Iraqi Kurdistan Region. Sustainability 2019, 11, (2), 485.

We have also concluded the shortcomings and lessons of the existing research at the end of the introduction. First, existing studies do not exclude the influence of the external environment. Second, current studies mostly use a single indicator to analyze the urbanization process, which cannot reflect the level of comprehensive urbanization and sustainable urbanization. Third, existing studies have not considered spatial differences in the policy effects of DZs. Therefore, we have made a targeted research design to fill the above gaps. The relevant modifications are as follows.

To sum up, existing studies do not exclude the external environment's influence when analyzing DZ's policy effects of DZs. The policy effects of DZs on urbanization have not been thoroughly examined. Second, existing studies mostly use a single indicator to analyze the urbanization process, which cannot reflect the level of comprehensive urbanization and sustainable urbanization. Third, current studies have not considered spatial differences in the policy effects of DZs. Therefore, this paper uses the DID model to quantify the role of DZs in China's urbanization. We conduct a series of robustness analyses to exclude the influence of external circumstances on the results. Considering that urbanization is a comprehensive process, we evaluate the spillover effects of DZs on urbanization from the perspectives of population, land, and the economy. In addition, this paper identifies the heterogeneous influence of DZs on urbanization to serve targeted urbanization development policies.

We have used English editing service to polish the manuscript and provided a polished manuscript as proof of editing. Once again, we appreciate sincerely for the editors/reviewers’ warm and patient work, and thank you very much for your constructive comments and suggestions which really promoted the quality of our manuscript. We hope that our earnest revision will conform to the approval of Land.

*********************************************************************

References

  1. Turok, I.; Mykhnenko, V., The trajectories of European cities, 1960–2005. Cities 2007, 24, (3), 165-182.
  2. Mulligan, G. F., Reprint of: Revisiting the urbanization curve. Cities 2013, 32, S58-S67.
  3. Stein, S., Capital city: Gentrification and the real estate state. Verso Books: 2019.
  4. Atkinson, R., Alpha city: How London was captured by the super-rich. Verso Books: 2021.
  5. Mueller, N.; Rojas-Rueda, D.; Khreis, H.; Cirach, M.; Andrés, D.; Ballester, J.; Bartoll, X.; Daher, C.; Deluca, A.; Echave, C., Changing the urban design of cities for health: The superblock model. Environment international 2020, 134, 105132.
  6. Todorov, V.; Dimov, I., Innovative digital stochastic methods for multidimensional sensitivity analysis in air pollution modelling. Mathematics 2022, 10, (12), 2146.
  7. Piracha, A.; Chaudhary, M. T., Urban air pollution, urban heat island and human health: A review of the literature. Sustainability 2022, 14, (15), 9234.
  8. Liu, H.; Li, Q.; Yu, D.; Gu, Y., Air quality index and air pollutant concentration prediction based on machine learning algorithms. Applied Sciences 2019, 9, (19), 4069.
  9. Jarah, S. H. A.; Zhou, B.; Abdullah, R. J.; Lu, Y.; Yu, W., Urbanization and urban sprawl issues in city structure: A case of the Sulaymaniah Iraqi Kurdistan Region. Sustainability 2019, 11, (2), 485.
  10. Chen, M.; Liu, W.; Tao, X., Evolution and assessment on China's urbanization 1960–2010: under-urbanization or over-urbanization? Habitat International 2013, 38, 25-33.
  11. Chen, M.; Liu, W.; Lu, D.; Chen, H.; Ye, C., Progress of China's new-type urbanization construction since 2014: A preliminary assessment. Cities 2018, 78, 180-193.
  12. Wang, X.-R.; Hui, E. C.-M.; Choguill, C.; Jia, S.-H., The new urbanization policy in China: Which way forward? Habitat International 2015, 47, 279-284.
  13. Bai, X.; Shi, P.; Liu, Y., Society: Realizing China's urban dream. Nature 2014, 509, (7499), 158-160.
  14. Chen, M.; Zhou, Y.; Huang, X.; Ye, C., The integration of new-type urbanization and rural revitalization strategies in China: origin, reality and future trends. Land 2021, 10, (2), 207.
  15. Guan, X.; Wei, H.; Lu, S.; Dai, Q.; Su, H., Assessment on the urbanization strategy in China: Achievements, challenges and reflections. Habitat International 2018, 71, 97-109.
  16. Feng, W.; Liu, Y.; Qu, L., Effect of land-centered urbanization on rural development: A regional analysis in China. Land Use Policy 2019, 87, 104072.
  17. Li, Y.; Li, Y.; Zhou, Y.; Shi, Y.; Zhu, X., Investigation of a coupling model of coordination between urbanization and the environment. Journal of environmental management 2012, 98, 127-133.
  18. Li, Q.; Chen, Y.; Liu, J., On the development mode of Chinese urbanization. Social Sciences in China 2012, 7, 82-100.
  19. Gao, S.; Wang, S.; Sun, D., Development zones and their surrounding host cities in China: Isolation and mutually beneficial interactions. Land 2021, 11, (1), 20.
  20. Chen, B.; Lu, M.; Timmins, C.; Xiang, K. Spatial misallocation: Evaluating place-based policies using a natural experiment in China; National Bureau of Economic Research: 2019.
  21. Wei, Y. D.; Leung, C. K., Development zones, foreign investment, and global city formation in Shanghai. Growth and Change 2005, 36, (1), 16-40.
  22. Cheng, H.; Liu, Y.; He, S.; Shaw, D., From development zones to edge urban areas in China: A case study of Nansha, Guangzhou City. Cities 2017, 71, 110-122.
  23. Jiang, Y.; Waley, P., Keeping up with the zones (es): how competing local governments in China use development zones as back doors to urbanization. Urban Geography 2022, 1-21.
  24. Schwarz, J. E.; Volgy, T. J., Experiments in employment-A British cure. Harvard Business Review 1988, 66, (2), 104-112.
  25. Erickson, R. A.; Syms, P. M., The effects of enterprise zones on local property markets. Regional Studies 1986, 20, (1), 1-14.
  26. Papke, L. E., Tax policy and urban development: evidence from the Indiana enterprise zone program. Journal of Public Economics 1994, 54, (1), 37-49.
  27. Bondonio, D.; Greenbaum, R. T., Do local tax incentives affect economic growth? What mean impacts miss in the analysis of enterprise zone policies. Regional science and urban economics 2007, 37, (1), 121-136.
  28. Greenbaum, R. T.; Engberg, J. B., The impact of state enterprise zones on urban manufacturing establishments. Journal of Policy Analysis and Management 2004, 23, (2), 315-339.
  29. Neumark, D.; Kolko, J., Do enterprise zones create jobs? Evidence from California’s enterprise zone program. Journal of Urban Economics 2010, 68, (1), 1-19.
  30. Begg, I., High technology location and the urban areas of Great Britain: developments in the 1980s. Urban studies 1991, 28, (6), 961-981.
  31. Gobillon, L.; Magnac, T.; Selod, H., Do unemployed workers benefit from enterprise zones? The French experience. Journal of Public Economics 2012, 96, (9-10), 881-892.
  32. Li, Y.; Wang, X., Innovation in suburban development zones: Evidence from Nanjing, China. Growth and Change 2019, 50, (1), 114-129.
  33. Wang, J., The economic impact of special economic zones: Evidence from Chinese municipalities. Journal of development economics 2013, 101, 133-147.
  34. Lu, Y.; Wang, J.; Zhu, L., Do place-based policies work? Micro-Level evidence from China's economic zone program. Micro-Level Evidence from China's Economic Zone Program (July 3, 2015) 2015.
  35. Sun, Y.; Ma, A.; Su, H.; Su, S.; Chen, F.; Wang, W.; Weng, M., Does the establishment of development zones really improve industrial land use efficiency? Implications for China’s high-quality development policy. Land Use Policy 2020, 90, 104265.
  36. Huang, Z.; He, C.; Wei, Y. D., A comparative study of land efficiency of electronics firms located within and outside development zones in Shanghai. Habitat International 2016, 56, 63-73.
  37. Wong, S.-W.; Tang, B.-s., Challenges to the sustainability of ‘development zones’: A case study of Guangzhou Development District, China. Cities 2005, 22, (4), 303-316.
  38. Cartier, C., 'Zone fever', the arable land debate, and real estate speculation: China's evolving land use regime and its geographical contradictions. Journal of Contemporary China 2001, 10, (28), 445-469.
  39. Zhang, J., Interjurisdictional competition for FDI: The case of China's “development zone fever”. Regional Science and Urban Economics 2011, 41, (2), 145-159.
  40. Cui, J.; Zhang, J.; Zheng, Y. In Carbon pricing induces innovation: evidence from China's regional carbon market pilots, AEA Papers and Proceedings, 2018; 2018; pp 453-57.
  41. Heckert, M.; Mennis, J., The economic impact of greening urban vacant land: a spatial difference-in-differences analysis. Environment and Planning A 2012, 44, (12), 3010-3027.
  42. Sivadasan, J., Barriers to competition and productivity: Evidence from India. The BE Journal of Economic Analysis & Policy 2009, 9, (1).
  43. Beck, T.; Levine, R.; Levkov, A., Big bad banks? The winners and losers from bank deregulation in the United States. The Journal of Finance 2010, 65, (5), 1637-1667.
  44. Wing, C.; Simon, K.; Bello-Gomez, R. A., Designing difference in difference studies: best practices for public health policy research. Annu Rev Public Health 2018, 39, (1), 453-469.
  45. Yu, B., Ecological effects of new-type urbanization in China. Renewable and Sustainable Energy Reviews 2021, 135, 110239.
  46. Lv, T.; Wang, L.; Zhang, X.; Xie, H.; Lu, H.; Li, H.; Liu, W.; Zhang, Y., Coupling coordinated development and exploring its influencing factors in Nanchang, China: From the perspectives of land urbanization and population urbanization. Land 2019, 8, (12), 178.
  47. Deng, X.; Huang, J.; Rozelle, S.; Zhang, J.; Li, Z., Impact of urbanization on cultivated land changes in China. Land use policy 2015, 45, 1-7.
  48. Zhang, H.; Chen, M.; Liang, C., Urbanization of county in China: Spatial patterns and influencing factors. Journal of Geographical Sciences 2022, 32, (7), 1241-1260.
  49. Chen, A.; Partridge, M. D., When are cities engines of growth in China? Spread and backwash effects across the urban hierarchy. Regional Studies 2013, 47, (8), 1313-1331.
  50. Kuang, W.; Liu, J.; Dong, J.; Chi, W.; Zhang, C., The rapid and massive urban and industrial land expansions in China between 1990 and 2010: A CLUD-based analysis of their trajectories, patterns, and drivers. Landscape and Urban Planning 2016, 145, 21-33.
  51. He, C.; Zhao, Y.; Huang, Q.; Zhang, Q.; Zhang, D., Alternative future analysis for assessing the potential impact of climate change on urban landscape dynamics. Science of the Total Environment 2015, 532, 48-60.
  52. Gebel, M.; Voßemer, J., The impact of employment transitions on health in Germany. A difference-in-differences propensity score matching approach. Social science & medicine 2014, 108, 128-136.
  53. Blundell, R.; Costa Dias, M., Evaluation methods for non‐experimental data. Fiscal studies 2000, 21, (4), 427-468.
  54. Rosenbaum, P. R.; Rubin, D. B., The central role of the propensity score in observational studies for causal effects. Biometrika 1983, 70, (1), 41-55.
  55. Gao, S.; Zhang, J.; Mo, X.; Wu, R., Dynamic Evolution of the Operating Efficiency of Development Zones in China. Sustainability 2021, 13, (18), 10395.

Reviewer 2 Report

The purpose of the authors is to take the panel data of 235 cities in China from 1990 to 2017 to evaluate the  policy effects of setting up development zones on urbanization from the perspectives of the population, land, and the economy. 

The topic is significant.  Submitted manuscript is written in high quality English language. Research questions are clearly formulated, so it is easy to understand the aim of this research. Methodology used in this research is appropriate. The manuscript is full of valuable Figures and Tables which clearly demonstrate the strength of the paper. However the formulas are not correctly displayed and need to be rewritten.

The formulas on line 99-101 need to be rewritten because they are displayed badly in the text, more explanation is needed.

I suggest slightly bigger description of the Figure 1 and Figure 2, it is difficult to read it.

On some places the quality of the English need to be improved.

When you talk about the development process of urbanization that invlolves other fields on line 122 and below I strongly suggest to say a few words about global health and ecological security and to cite these papers in the field:

Todorov, V.; Dimov, I. Innovative Digital Stochastic Methods for Multidimensional Sensitivity Analysis in Air Pollution Modelling. Mathematics 2022, 10, 2146. https://doi.org/10.3390/math10122146

Piracha, A.; Chaudhary, M.T. Urban Air Pollution, Urban Heat Island and Human Health: A Review of the Literature. Sustainability 2022, 14, 9234. https://doi.org/10.3390/su14159234

Liu, H.; Li, Q.; Yu, D.; Gu, Y. Air Quality Index and Air Pollutant Concentration Prediction Based on Machine Learning Algorithms. Appl. Sci. 2019, 9, 4069. https://doi.org/10.3390/app9194069

The weak point of this article is precisely the way it is written, which jeopardizes its following by readers, especially in a text of this complexity. A careful revision of the text is suggested, in order to use shorter sentences (some of the current ones should be divided), use commas and periods judiciously to clearly separate the ideas presented, and pay some attention to grammatical details.

I strongly suggest two separate sections Discussion and Conclusion, and more focus on Discussion section, Section 4 in the current form is unacceptable!

 You should add in the Conclusion about the novelties of the originality of the observed results mentioned already in the Discussion section. A few words about future work and to whom the survey will be important will be valuable.

I suggest to improve the text, update the References, write a separate Discussion section and improving the Conclusion and accepting after this.

Author Response

(The authors gave the same response as above.)

Reviewer 3 Report

The paper aims to assess the impact of establishing development zones on urbanization development using a static method known as difference-in-differences.

It is a good paper and it is based on a solid statistical approach but in my vision it is lacking in reflection. I think it could appears more interesting for a researcher in economics than in urban planning.

Furthermore, authors focus on quantifying impacts of DZ zones but a not considered question is: what about the quality of the impact?

In any case, I think some improvements can be made.

Abstract.

In row 11 authors wrote: “The sustainable development of urbanization is a necessary condition for China to realize modernization”. But the analysis made is not focused on sustainability. Sustainability is nowhere else mentioned in the paper, and there is no a definition of it.

Introduction.

The DZ model is the core of the paper, so I suggest looking into the subject in more depth. More information would be appreciated (for example what purpose they serve, through which policies they were introduced and when, how the areas were selected, what criteria were used).

In row 80 authors wrote: “Therefore, this paper uses the DID model to quantify the role of DZs in China's urbanization to fill the above research gaps”. I suggest making it explicitly clear what gaps are intended to be bridged by the analysis”.

Method:

In row 89 authors wrote “using the DID method”. The acronym is introduced without giving a definition.

In row 91 authors wrote “from the perspectives of the population, land, and the economy”. It seems a too general sentence, I suggest clarifying which aspects of these elements (population, land, economy) authors intend to investigate.

Please clarify why the DID model was chosen and how it works from a conceptual point of view (for those who are not statisticians).

Why are the 'propensity score matching' and 'kernel matching method' not mentioned in this paragraph on the method?

Furthermore, measuring social development solely by a purely economic parameter/s does not seem sufficient.

Conclusion.

There is no critical mention of the fact that the effects of urbanisation should also include issues such as the environment, quality of life, social relations, services, land consumption, climate effects, equity, etc. in their assessment.

In row 346 authors wrote: “High-quality urbanization is an important support mechanism for China's sustainable development”.

However I think that the parameters used are not adequate to define quality. So what do you mean by “high quality urbanization”?

Author Response

5-September-2022

Dear Editor,

We are very grateful for having a chance to improve our manuscript “The impact of development zones on China's urbanization: Perspectives from population, land, and the economy” (Land-1889747). We also appreciate the editors and reviewers who reviewed our research and paid so much patience to our manuscript; the detailed comments and suggestions are very significant and helpful for the authors to improve the research.

Based on the comments and suggestions, we have made careful modifications to the manuscript. The main corrections in the paper and the responds to the reviewers’ comments are appended below. The detailed information can also be seen in our revised manuscript. Revised portions are marked in red color in the revised paper.

Although the authors have carefully improved the paper in accordance with the comments and suggestions, there may still exist some problems and errors in our revised manuscript. We invite the editors and referees to propose more criticisms and suggestions. We also hope the new manuscript will meet Land’s standard with approval.

Best regards.

Yours sincerely,

The Authors.

Response to Reviewers
Land-1889747
The impact of development zones on China's urbanization: Perspectives from population, land, and the economy

1. Response to Reviewer #1:
(1) The topic feels incomplete. For example, how does land affect urbanization? If this paper is to investigate land effects on urbanization, it should be appropriate to use a spatial econometric model.
Response: Thank you very much for your reminder. The purpose of this paper is to explore whether the establishment of DZs affects urbanization, whether this effect is different among regions. Previous studies have not made an in-depth analysis of this topic. The results of this paper confirm that the establishment of DZs has a significant impact on urbanization. Based on this, we believe that it is vital and appropriate to continue to explore how DZs affect urbanization, which is also our future research direction. According to your reminder, we have added future research orientation in the discussion section. The additions are as follows.
In addition, another limitation of this paper was that it only focused on the correlation between urbanization and DZ policy without an in-depth discussion on the mechanism. Based on this study, it might prove fruitful to investigate further the impact mechanism of DZs on urbanization from the perspectives of environment, quality of life, social relations, services, and equity.

(2) The limitations of the study should also be elaborated. In particular, the meaning of policies should be analyzed in depth.
Response: Thank you very much for your reminder. According to your reminder, we have added the innovations and significance of this study in the conclusion section. In addition, we have also elaborated limitations of this paper and proposed our future research prospects. The relevant modifications are as follows.
This study has provided empirical findings which deepen the understanding of the link between the urbanization process and DZ policy. These findings provide a vital policy reference for regions in formulating urbanization strategies. Compared with the existing research, the design of this paper has the following innovations. First, this paper uses the DID model to explore the impact of the DZ policy on urbanization. It conducts propensity score matching analysis and parallel trend test on the research samples. The results of the study exclude the influence of exogenous factors and have strong credibility. In addition, this paper investigates the urbanization process from the perspectives of population, land, and economy, which can reflect the comprehensive urbanization level. We believe this has important implications for identifying the sustainability of urbanization. Since there is no authoritative definition of sustainable urbanization, although this paper tries to use multiple urbanization indicators, it still cannot fully reflect sustainable urbanization. In the future, more in-depth research on sustainable urbanization should be carried out based on sustainable development. In addition, another limitation of this paper was that it only focused on the correlation between urbanization and DZ policy without an in-depth discussion on the mechanism. Based on this study, it might prove fruitful to investigate further the impact mechanism of DZs on urbanization from the perspectives of environment, quality of life, social relations, services, and equity.

(3) The literature review is chaotic, and it is difficult for readers to understand its context. Many sentences have obvious non-native English expressions.
Response: Thank you very much for your reminder. We have reorganized the introduction part to be more logical. In the introduction, the first and second paragraphs respectively introduce the research status of world urbanization and China's urbanization and emphasized the importance of sustainable urbanization research. The third, fourth, and fifth paragraphs introduce the DZ model of urbanization and do a literature review on the policy effects of the DZ. The sixth paragraph clarifies the objective of this study, or the main innovation of this study. The related modifications of the introduction are as follows.
The emergence of cities is a sign of human maturity and civilization [1]. After two centuries of unprecedented rapid urbanization, more than half of the world's population lives in cities [2]. Rapid urbanization has a profound impact on global sustainability while increasing social productivity. The research on urbanization began at the beginning of the emergence of modern industrial cities and has been widely concerned by the academic community. Scholars believe that while urbanization promotes social modernization in various countries, it also brings a series of problems. For example, studies in London and New York found that state capital and planning play an essential role in urbanization [3]. In urban renewal, land values and rents have risen rapidly. However, ordinary citizens do not benefit from it, and the the wealth gap problem becomes more significant in urbanization [4]. The rapid development of urbanization has also deepened environmental issues and aroused academic attention to healthy and ecological cities [5]. Scholars have deeply analyzed urban air pollution [6], heat island effect [7], ecological pattern [8], land expansion [9], and other issues under the process of urbanization, and explored sustainable urbanization models. 
Since 1990, China's urbanization has entered a stage of rapid development [10]. Urbanization drives not only large-scale population agglomeration to cities [11], but also promotes the urban spatial restructuring and industrial restructuring [12]. Sustainable urbanization is an important driving force and necessary condition for national modernization [13, 14]. For China, guiding sustainable urbanization not only determines the future of China's urbanization but also affects the prospects of world urbanization [15]. Therefore, it is crucial to explore the sustainable urbanization model. The academic community has not formed an authoritative definition of sustainable urbanization. With in-depth research on the process of urbanization in different disciplines, the concept of urbanization has also been extended [16]. Urbanization has gradually become a comprehensive process involving population migration, industrial structure adjustment, and land-use patterns transformation [17]. It is of great significance to identify the sustainable urbanization model from population, industry, and land perspectives. 
Some scholars have summarized China's urbanization development into seven "promoting models". Among these, the "development zone (DZ) model" is one of the most representative [18]. The " DZ model" is a typical example of government-led urbanization. The government establishes DZ in regions with advantageous locations and provides a series of preferential policies [19]. China's first batch of DZs was all set up in eastern port cities with advantageous locations. To support the development of the DZs, the State gives preferential policies such as tax reduction and financial support. The DZ can accept the transfer of international capital and industries by its policy superiority and favorable investment environment [20]. Since establishing the first national DZ in 1984, DZs have become an important strategic initiative to promote local economic growth [21]. As of 2018, China has established a total of 552 national DZs. The location of DZs has also shifted from the eastern coastal regions to the relatively backward central and western regions. By establishing DZs, cities can complete industrial and population agglomeration [22], and achieve transformation of land structure and industrial structure [23]. Due to the superiority of the "DZ model", it is necessary to estimate the impact of DZs on urbanization.
Existing studies are less concerned with the impact of DZs on urbanization. The related research on DZs and urban economic development, land use, and labor markets provide a new perspective. British scholars found that establishing industrial space policies similar to DZs did not have an apparent driving effect on regional economic growth [24]. In contrast, the employment costs within the DZs were remarkably increased [25]. In addition, studies on DZs in the United States have found that although enterprises in DZs can enjoy tax deductions, employees' wages are lower [26]. Besides economic growth, the driving effect of DZs construction on regional employment has also attracted the attention of scholars [27]. Likewise, most studies on the labor market of DZs in the United States do not yield positive conclusions [28, 29]. However, research on the DZ policy in European countries has found that the construction of DZs can attract large numbers of the employed population [30]. At the same time, the regional high-tech industries have also been significantly developed [31].
For China, many studies have confirmed the positive effects of the DZ policy and promoted urban development through the following aspects. First, the establishment of the DZ is accompanied by large-scale infrastructure construction, which improves the spatial appearance of the city [32]. Second, the DZ attracts foreign investment into the city through preferential policies, forming industrial agglomerations and becoming the growth pole of the city [33]. Third, enterprises settled in the DZ increased not only urban tax revenue but also attracted labor accumulation [34]. Finally, DZs can increase urban productivity [35] and land-use efficiency [36] through their scale effect. Nevertheless, the development of China's DZs has not always been smooth. On the one hand, due to the excessive proportion of foreign investment in the DZs, they are affected by the international economic cycle. On the other hand, the abuse of preferential policies also leads to the spillover effects of DZs is no longer obvious [37]. Even between 1990 and 2005, due to the blind establishment of DZs by local governments, the phenomenon of "zone fever" frequently occurred in China [38]. The positive effect of the DZ is no longer significant. [39].

(4) References cannot be displayed in a stacked fashion. For example. “Likewise, most studies on the labor market of DZs in the United 56 States do not yield positive conclusions [18-20]”. 
Response: Thank you very much for your suggestion. According to your suggestion, we have separated the quotes references. The relevant modifications are as follows.
Since 1990, China's urbanization has entered a stage of rapid development [10]. Urbanization drives not only large-scale population agglomeration to cities [11], but also promotes urban spatial restructuring and industrial restructuring [12].
With the in-depth research on the process of urbanization in different disciplines, the concept of urbanization has also been extended [16]. Urbanization has gradually become a comprehensive process involving population migration, adjustment of industrial structure, and transformation of land-use patterns [17].
The government establishes DZ in regions with advantageous locations and provides a series of preferential policies [19]. The DZ can accept the transfer of international capital and industries by virtue of its policy superiority and favorable investment environment [20].
Through the establishment of DZs, cities can complete industrial and population agglomeration [22], and achieve transformation of land structure and industrial structure [23].
British scholars found that establishing industrial space policies similar to DZs did not have an obvious driving effect on regional economic growth [24]. In contrast, the employment costs within the DZs were greatly increased [25].
However, research on the DZ policy in European countries has found that the construction of DZs can attract large numbers of the employed population [30], and significantly promote the development of regional high-tech industries [31].
Finally, DZs can increase urban productivity [35] and land-use efficiency [36] through their scale effect.
Even between 1990 and 2005, due to the blind establishment of DZs by local governments, the phenomenon of "zone fever" frequently occurred in China [38], which greatly reduced the positive impact of DZs [39].

(5) Introduction: The objective of the paper presented need more clarifications to suit reader to understand the main idea of the paper. The research purpose and contributions of the article are unclear.
Response: Thank you very much for your suggestion. The purpose of this paper is to explore China's sustainable urbanization model, which determines the future development prospects of China and even the world. Through analysis, we find that DZs play a significant role in the urbanization process, especially sustainable urbanization. Due to the superiority of the "DZ model", it is necessary to estimate the impact of DZs on urbanization. In addition, it is of great significance to identify the sustainable urbanization model from the perspectives of population, economy, and land. Therefore, this paper uses the DID model to quantify the role of DZs in China's urbanization from perspectives of population, land, and economy. According to your suggestion, we have summarized the shortcomings of existing research in the introduction. The research purpose and contributions of the article have been put forward. The relevant modifications are as follows.
To sum up, existing studies do not exclude the external environment's influence when analyzing DZ's policy effects of DZs. The policy effects of DZs on urbanization have not been thoroughly examined. Second, existing studies mostly use a single indicator to analyze the urbanization process, which cannot reflect the level of comprehensive urbanization and sustainable urbanization. Third, current studies have not considered spatial differences in the policy effects of DZs. Therefore, this paper uses the DID model to quantify the role of DZs in China's urbanization. We conduct a series of robustness analyses to exclude the influence of external circumstances on the results. Considering that urbanization is a comprehensive process, we evaluate the spillover effects of DZs on urbanization from the perspectives of population, land, and the economy. In addition, this paper identifies the heterogeneous influence of DZs on urbanization to serve targeted urbanization development policies.

(6) The discussion of empirical results is too brief. Why does this article have no policy recommendations?
Response: Thank you very much for your suggestion. According to your suggestion, we have reorganized and separated the conclusion and discussion sections of this article. We have also expanded the discussion section and put forward three policy suggestions for the planning and construction of the DZ. First, continue to promote the construction of DZs in various regions, and give full play to the driving role of DZs in urbanization. Second, change the development mode of the DZ and improve the construction level of its supporting facilities. Third, strengthen the spillover effect of DZs on the urbanization of surrounding areas. The relevant modifications are as follows.
4. Discussion
China has entered the "new normal" stage of economic development. Sustainable urbanization is essential to support China's economic transformation. Urbanization promoted by DZs not only leads to population agglomeration in cities but also changes cities' land use and industrial structure. This paper takes national DZs as the research object and empirically analyzes the impact of DZs on urbanization. Specifically, we collected the DZs and urbanization data of 235 cities in China from 1990 to 2017 and constructed a DID model to carry out the study. Urbanization is a complex and comprehensive process. PU can reflect population agglomeration. LU can reflect the expansion of urban construction land. EU can reflect the contribution of secondary and tertiary industries to economic development. Urbanization is highly sustainable with the support of population, infrastructure, and industries. Therefore, this paper examines the impact of DZs on urbanization from the perspectives of population, land, and economy, which is of great significance for judging the sustainable development of urbanization. In addition, China's regional development is significantly unbalanced. The spillover effects of DZs on urbanization are also different among regions. However, less attention has been paid to this issue in existing studies. This paper divides the sample into three regions, namely the eastern, central, and western regions, to identify the differences among policy effects in different regions of China.
The results show that the DZ policy benefited the intensive use of construction land and the upgrading of the industrial structure to a certain extent. The regional regression results show that the impact of DZs on urbanization presented significant heterogeneity. The results of this study support that the advantage of the DZ policy for the urbaniza-tion of the eastern region was to strengthen the intensive use of construction land. For the central and western regions with relatively weak foundational development, DZs had played a vital role in attracting population and upgrading industries.  However, DZs also increased the area of construction land in central and western cities, which may be detrimental to the intensive use of urban construction land.
Although the above enlightenment is only a preliminary conclusion, it has essential reference value for the planning and construction of the DZ. First, continue to promote the construction of DZs in various regions and give full play to the driving role of DZs in urbanization. The establishment of DZs shall be subject to strict examination and approval management, and the development zones with low developmental benefits and inconsistent standards shall be addressed. In contrast, maintaining the quantity and quality of the DZ, strengthening its attraction to the population, and promoting its intensive land use and up-grading of industry. Second, change the development mode of the DZ and improve the construction level of its supporting facilities. The DZ is regarded not only as an industrial agglomeration area but is built into a new urban district that complements the host city. By improving infrastructure construction and optimizing the living environment, the DZ will attract population gathering and promote the development of urbanization. For example, enhance public service facilities such as sports, medical care, and education in the DZ and improve its living functions. Third, strengthen the spillover effect of DZs on the urbanization of surrounding areas. View the DZ as a link between the city center and the suburbs. Strengthen the cooperation of transportation infrastructure between the DZ and its surrounding areas. By constructing a regionally integrated transportation network, the radiation effect of the DZ on the surrounding areas will be enhanced. It is of great significance to drive the urbanization development of the surrounding areas of the DZ.
5. Conclusion
Using the DID model, this paper focuses on analyzing the spillover effect of the DZ policy on urbanization. We found that after conducting propensity score matching, a parallel trend test, and controlling a series of variables, the DZ policy had no significant impact on PU. The establishment of DZs across the country had reduced the LU level in their host cities by about 22.6%, while the EU level had increased by about 3.2%. For the eastern region, the DZ policies had no significant impact on PU but negatively impacted LU and EU. The LU level of the cities with DZs was reduced by 6%, and the EU level was decreased by 1.2%. For the central and western regions with relatively weak foundational development, DZs played a vital role in attracting the population and upgrading industries. However, DZs also increased the area of construction land in central and western cities, which may be detrimental to the intensive use of urban construction land.
This study has provided empirical findings which deepen the understanding of the link between the urbanization process and DZ policy. These findings provide a vital policy reference for regions in formulating urbanization strategies. Compared with the existing research, the design of this paper has the following innovations. First, this paper uses the DID model to explore the impact of the DZ policy on urbanization. It conducts propensity score matching analysis and parallel trend test on the research samples. The results of the study exclude the influence of exogenous factors and have strong credibility. In addition, this paper investigates the urbanization process from the perspectives of population, land, and economy, which can reflect the comprehensive urbanization level. We believe this has important implications for identifying the sustainability of urbanization. Since there is no authoritative definition of sustainable urbanization, although this paper tries to use multiple urbanization indicators, it still cannot fully reflect sustainable urbanization. In the future, more in-depth research on sustainable urbanization should be carried out based on sustainable development. In addition, another limitation of this paper was that it only focused on the correlation between urbanization and DZ policy without an in-depth discussion on the mechanism. Based on this study, it might prove fruitful to investigate further the impact mechanism of DZs on urbanization from the perspectives of environment, quality of life, social relations, services, and equity.

(7) There are too many basic grammar issues, it is better to spend more time on polishing the writings. 
Response: Thank you very much for your reminder. We have polished the writings and checked the text for possible grammar issues. We have used English editing service to polish the manuscript and provided a polished manuscript as proof of editing.

2. Response to Reviewer #2:
The purpose of the authors is to take the panel data of 235 cities in China from 1990 to 2017 to evaluate the policy effects of setting up development zones on urbanization from the perspectives of the population, land, and the economy. The topic is significant. Submitted manuscript is written in high quality English language. Research questions are clearly formulated, so it is easy to understand the aim of this research. Methodology used in this research is appropriate. The manuscript is full of valuable Figures and Tables which clearly demonstrate the strength of the paper. I suggest to improve the text, update the References, write a separate Discussion section and improving the Conclusion and accepting after this.
(1) The formulas are not correctly displayed and need to be rewritten. The formulas on line 99-101 need to be rewritten because they are displayed badly in the text, more explanation is needed.
Response: Thank you very much for your suggestion. This paper uses the DID model to evaluate the impact of DZ on urbanization. There are three dependent variables in this article. Therefore, we use three formulas to express the DID model. These formulas are the same except for dependent variables. This article does not explain the elements of each formula separately, which can cause formulas to display badly in the text. According to your suggestion, we combine the formulas of line 99-101 and line 213-215 to facilitate readers to better understanding. The modified formulas and explanations are as follows.
lnU_it=α_0+α_1 〖DZ〗_it+α_2 〖lnZ〗_it+δ_t+γ_i+ε_it,                     (1)
where U_it is the dependent variable, which represents the population urbanization (PU), land urbanization (LU), and economic urbanization (EU) levels of city i at time t, respectively; 〖DZ〗_it is the independent variable, which indicates whether city i has set up a DZ at time t; Z_it  represents the control variables such as per capita GDP (PcGDP), average wage (AS), budgetary revenue (BR), fixed asset in-vestment (FAI), foreign direct investment (FDI), political status (PS), nearshore distance (ND), and latitude (LA); α_0, α_1, and α_2 are regression coefficients, where α_1 is the focus of this paper and represents the impact of DZ policy on urbanization; δ_t and γ_i are the fixed effects on the time and individual, respectively; and ε_it is a random disturbance term. In order to eliminate the possible heteroscedasticity in the model, the dependent variables and the control variable are processed logarithmically in this paper. In addition, this paper divides the sample into three regions, namely the eastern, central, and western regions, to identify the differences among policy effects in different regions of China.
lnU_it=α_0+α_1 〖DZ〗_it^(-5)+α_2 〖DZ〗_it^(-4)+⋯+α_10 〖DZ〗_it^4++α_11 〖DZ〗_it^5 〖+δ〗_t+γ_i+ε_it,     (2)
where U_it is the dependent variable, which represents the population urbanization (PU), land urbanization (LU), and economic urbanization (EU) levels of city i at time t, respectively; where 〖DZ〗_it is the dummy variable for establishing the DZ; 〖DZ〗_it^(-5) indicates that time t is 5 years before the establishment of the DZ in city i, it is set as 1; otherwise, it is set as 0; 〖DZ〗_it^5 indicates that if time t is 5 years after the establishment of the DZ in city i, it is set as 1; otherwise, it is set as 0; and other dummy variables are defined similarly; δ_t and γ_i are the fixed effects on time and the individual, respectively; and ε_it is a random disturbance term.

(2) I suggest slightly bigger description of the Figure 1 and Figure 2, it is difficult to read it.
Response: Thank you very much for your suggestion. According to your suggestion, we have added descriptions of Figure 1 and Figure 2. The additions are as follows.
Figure 1 displays the spatial pattern of the PU, LU, and EU levels for the treatment and control groups in 1990 and 2017. From Figure 1, the PU, LU, and EU levels varied significantly among cities. The difference in LU level between cities was the most significant, followed by PU. And the EU level was relatively balanced among cities. These reflected the complexity of the urbanization process. Therefore, it was necessary to measure the impact of DZs on urbanization from different perspectives. On the whole, the urbanization level in the eastern coastal regions was observably higher than that in the central and western regions. The Beijing-Tianjin-Hebei Urban Agglomerations, Yangtze River Delta Urban Agglomerations, and Pearl River Delta Urban Agglomerations were the high-value areas of urbanization level, and the increase was the most obvious during the study period. The regional difference in urbanization level proved that it was reasonable to divide the sample into three regions. In addition, Figure 1 further confirmed that the urbanization level of the treatment group cities was higher than that of the control group.

Figure 1. Urbanization spatial pattern of treatment group and control group in China
As can be seen from Figure 2, the establishment of DZs exerted an impact on urbanization. And the impact of DZs on urbanization increases significantly over time. The policy effect on urbanization in years without DZs fluctuated around the zero value. This indicated no unparallel trend in the sample before the city set up the DZ. However, after the establishment of DZs, the policy effect of PU, LU, and EU significantly deviated from a zero value. It showed that the establishment of DZs exerted an impact on urbanization progress. And the treatment and control groups in this paper also passed the parallel trend test. The establishment of DZs in cities significantly affected the inherent processes of PU, LU, and EU. Specifically, the DZ policy exerted an incredibly positive effect on PU and EU, while it negatively affected LU. In addition, Figure 2 showed that the policy effect of DZs on urbanization increases significantly over time. Because of this, we have reason to believe that establishing DZs can significantly change the developmental trend of the urban population, land use, and industrial industry, and these impacts will become more pronounced over time.

Figure 2. Parallel trend test plots

(3) On some places the quality of the English need to be improved. 
Response: Thank you very much for your reminder. We have polished the writings and checked the text for possible grammar issues. We have used English editing service to polish the manuscript and provided a polished manuscript as proof of editing.

(4) When you talk about the development process of urbanization that invlolves other fields on line 122 and below I strongly suggest to say a few words about global health and ecological security and to cite these papers in the field:
 Todorov, V.; Dimov, I. Innovative Digital Stochastic Methods for Multidimensional Sensitivity Analysis in Air Pollution Modelling. Mathematics 2022, 10, 2146. https://doi.org/10.3390/math10122146
 Piracha, A.; Chaudhary, M.T. Urban Air Pollution, Urban Heat Island and Human Health: A Review of the Literature. Sustainability 2022, 14, 9234. https://doi.org/10.3390/su14159234
 Liu, H.; Li, Q.; Yu, D.; Gu, Y. Air Quality Index and Air Pollutant Concentration Prediction Based on Machine Learning Algorithms. Appl. Sci. 2019, 9, 4069. https://doi.org/10.3390/app9194069
Response: Thank you very much for your suggestion. According to your suggestion, we have discussed global health and ecological security issues in the process of urbanization in the introduction. In addition, we have added the discussion on urban capital and land sprawl under the process of urbanization, citing international literature in these fields. The additions are as follows.
The emergence of cities is a sign of human maturity and civilization [1]. After two centuries of unprecedented rapid urbanization, more than half of the world's population lives in cities [2]. Rapid urbanization has a profound impact on global sustainability while increasing social productivity. The research on urbanization began at the beginning of the emergence of modern industrial cities and has been widely concerned by the academic community. Scholars believe that while urbanization promotes social modernization in various countries, it also brings a series of problems. For example, studies in London and New York found that state capital and planning play an essential role in urbanization [3]. In urban renewal, land values and rents have risen rapidly. However, ordinary citizens do not benefit from it, and the the wealth gap problem becomes more significant in urbanization [4]. The rapid development of urbanization has also deepened environmental issues and aroused academic attention to healthy and ecological cities [5]. Scholars have deeply analyzed urban air pollution [6], heat island effect [7], ecological pattern [8], land expansion [9], and other issues under the process of urbanization, and explored sustainable urbanization models.
References:
1. Turok, I.; Mykhnenko, V., The trajectories of European cities, 1960–2005. Cities 2007, 24, (3), 165-182.
2. Mulligan, G. F., Reprint of: Revisiting the urbanization curve. Cities 2013, 32, S58-S67.
3. Stein, S., Capital city: Gentrification and the real estate state. Verso Books: 2019.
4. Atkinson, R., Alpha city: How London was captured by the super-rich. Verso Books: 2021.
5. Mueller, N.; Rojas-Rueda, D.; Khreis, H.; Cirach, M.; Andrés, D.; Ballester, J.; Bartoll, X.; Daher, C.; Deluca, A.; Echave, C., Changing the urban design of cities for health: The superblock model. Environment international 2020, 134, 105132.
6. Todorov, V.; Dimov, I., Innovative digital stochastic methods for multidimensional sensitivity analysis in air pollution modelling. Mathematics 2022, 10, (12), 2146.
7. Piracha, A.; Chaudhary, M. T., Urban air pollution, urban heat island and human health: A review of the literature. Sustainability 2022, 14, (15), 9234.
8. Liu, H.; Li, Q.; Yu, D.; Gu, Y., Air quality index and air pollutant concentration prediction based on machine learning algorithms. Applied Sciences 2019, 9, (19), 4069.
9. Jarah, S. H. A.; Zhou, B.; Abdullah, R. J.; Lu, Y.; Yu, W., Urbanization and urban sprawl issues in city structure: A case of the Sulaymaniah Iraqi Kurdistan Region. Sustainability 2019, 11, (2), 485.

(5) The weak point of this article is precisely the way it is written, which jeopardizes its following by readers, especially in a text of this complexity. A careful revision of the text is suggested, in order to use shorter sentences (some of the current ones should be divided), use commas and periods judiciously to clearly separate the ideas presented, and pay some attention to grammatical details.
Response: Thank you very much for your suggestion. According to your suggestion, we have revised the text carefully. We have broken down some long sentences into shorter sentences. In addition, we have also checked the text for possible grammar issues. We have used English editing service to polish the manuscript and provided a polished manuscript as proof of editing.

(6) I strongly suggest two separate sections Discussion and Conclusion, and more focus on Discussion section, Section 4 in the current form is unacceptable!
Response: Thank you very much for your suggestion. According to your suggestion, we have reorganized and separated this article's conclusion and discussion section. In the discussion section, we focus on the purpose and significance of this study and put forward three policy suggestions for the planning and construction of the DZ. First, continue to promote the construction of DZs in various regions and give full play to the driving role of DZs in urbanization. Second, change the DZ's development mode and improve its supporting facilities' construction level. Third, strengthen the spillover effect of DZs on the urbanization of surrounding areas. The relevant modifications are as follows.
4. Discussion
China has entered the "new normal" stage of economic development. Sustainable urbanization is essential to support China's economic transformation. Urbanization promoted by DZs not only leads to population agglomeration in cities but also changes cities' land use and industrial structure. This paper takes national DZs as the research object and empirically analyzes the impact of DZs on urbanization. Specifically, we collected the DZs and urbanization data of 235 cities in China from 1990 to 2017 and constructed a DID model to carry out the study. Urbanization is a complex and comprehensive process. PU can reflect population agglomeration. LU can reflect the expansion of urban construction land. EU can reflect the contribution of secondary and tertiary industries to economic development. Urbanization is highly sustainable with the support of population, infrastructure, and industries. Therefore, this paper examines the impact of DZs on urbanization from the perspectives of population, land, and economy, which is of great significance for judging the sustainable development of urbanization. In addition, China's regional development is significantly unbalanced. The spillover effects of DZs on urbanization are also different among regions. However, less attention has been paid to this issue in existing studies. This paper divides the sample into three regions, namely the eastern, central, and western regions, to identify the differences among policy effects in different regions of China.
The results show that the DZ policy benefited the intensive use of construction land and the upgrading of the industrial structure to a certain extent. The regional regression results show that the impact of DZs on urbanization presented significant heterogeneity. The results of this study support that the advantage of the DZ policy for the urbaniza-tion of the eastern region was to strengthen the intensive use of construction land. For the central and western regions with relatively weak foundational development, DZs had played a vital role in attracting population and upgrading industries.  However, DZs also increased the area of construction land in central and western cities, which may be detrimental to the intensive use of urban construction land.
Although the above enlightenment is only a preliminary conclusion, it has essential reference value for the planning and construction of the DZ. First, continue to promote the construction of DZs in various regions and give full play to the driving role of DZs in urbanization. The establishment of DZs shall be subject to strict examination and approval management, and the development zones with low developmental benefits and inconsistent standards shall be addressed. In contrast, maintaining the quantity and quality of the DZ, strengthening its attraction to the population, and promoting its intensive land use and up-grading of industry. Second, change the development mode of the DZ and improve the construction level of its supporting facilities. The DZ is regarded not only as an industrial agglomeration area but is built into a new urban district that complements the host city. By improving infrastructure construction and optimizing the living environment, the DZ will attract population gathering and promote the development of urbanization. For example, enhance public service facilities such as sports, medical care, and education in the DZ and improve its living functions. Third, strengthen the spillover effect of DZs on the urbanization of surrounding areas. View the DZ as a link between the city center and the suburbs. Strengthen the cooperation of transportation infrastructure between the DZ and its surrounding areas. By constructing a regionally integrated transportation network, the radiation effect of the DZ on the surrounding areas will be enhanced. It is of great significance to drive the urbanization development of the surrounding areas of the DZ.

(7) You should add in the Conclusion about the novelties of the originality of the observed results mentioned already in the Discussion section. A few words about future work and to whom the survey will be important will be valuable.
Response: Thank you very much for your reminder. According to your reminder, we have added the innovations and significance of this study in the conclusion section. In addition, we have also elaborated limitations of this paper and proposed our future research prospects. The relevant modifications are as follows.
5. Conclusion
Using the DID model, this paper focuses on analyzing the spillover effect of the DZ policy on urbanization. We found that after conducting propensity score matching, a parallel trend test, and controlling a series of variables, the DZ policy had no significant impact on PU. The establishment of DZs across the country had reduced the LU level in their host cities by about 22.6%, while the EU level had increased by about 3.2%. For the eastern region, the DZ policies had no significant impact on PU but negatively impacted LU and EU. The LU level of the cities with DZs was reduced by 6%, and the EU level was decreased by 1.2%. For the central and western regions with relatively weak foundational development, DZs played a vital role in attracting the population and upgrading industries. However, DZs also increased the area of construction land in central and western cities, which may be detrimental to the intensive use of urban construction land.
This study has provided empirical findings which deepen the understanding of the link between the urbanization process and DZ policy. These findings provide a vital policy reference for regions in formulating urbanization strategies. Compared with the existing research, the design of this paper has the following innovations. First, this paper uses the DID model to explore the impact of the DZ policy on urbanization. It conducts propensity score matching analysis and parallel trend test on the research samples. The results of the study exclude the influence of exogenous factors and have strong credibility. In addition, this paper investigates the urbanization process from the perspectives of population, land, and economy, which can reflect the comprehensive urbanization level. We believe this has important implications for identifying the sustainability of urbanization. Since there is no authoritative definition of sustainable urbanization, although this paper tries to use multiple urbanization indicators, it still cannot fully reflect sustainable urbanization. In the future, more in-depth research on sustainable urbanization should be carried out based on sustainable development. In addition, another limitation of this paper was that it only focused on the correlation between urbanization and DZ policy without an in-depth discussion on the mechanism. Based on this study, it might prove fruitful to investigate further the impact mechanism of DZs on urbanization from the perspectives of environment, quality of life, social relations, services, and equity.

3. Response to Reviewer #3:
The paper aims to assess the impact of establishing development zones on urbanization development using a static method known as difference-in-differences. It is a good paper and it is based on a solid statistical approach but in my vision it is lacking in reflection. I think it could appears more interesting for a researcher in economics than in urban planning. Furthermore, authors focus on quantifying impacts of DZ zones but a not considered question is: what about the quality of the impact? In any case, I think some improvements can be made.
(1) In row 11 authors wrote: “The sustainable development of urbanization is a necessary condition for China to realize modernization”. But the analysis made is not focused on sustainability. Sustainability is nowhere else mentioned in the paper, and there is no a definition of it. 
Response: Thank you very much for your reminder. Although this paper does not focus on "sustainability", the topic and content of the study are all related to sustainable urbanization. On the one hand, the dependent variable of this paper is not only population urbanization (PU), but also land urbanization (LU) and economic urbanization (EU). LU can reflect the expansion of urban construction land. EU can reflect the contribution of secondary and tertiary industries to economic development. With the support of the population, infrastructure, and industries, urbanization is highly sustainable. Therefore, we believe that multiple perspectives can reflect the sustainability of urbanization development to a certain extent. On the other hand, the independent variable of this paper is the DZ policy. Previous studies have confirmed that urbanization development promoted by DZs has strong sustainability ("Li, Q.; Chen, Y.; Liu, J. On the development mode of Chinese urbanization. Social Sciences in China 2012, 7, 82-100 "). Because the DZ is the regional industrial agglomeration center and the growth pole of economic development, it has strong development vitality. The empirical results of this paper show that the DZ does not positively impact the PU, LU, and EU in all regions. Therefore, this paper puts forward policy suggestions to strengthen the role of DZ in promoting urbanization to realize "sustainable" urbanization. Due to the lack of definitions and standards of sustainable urbanization, the "sustainability" analysis of urbanization in this paper is still insufficient. According to your reminder, we take the analysis of sustainable urbanization as the research deficiency and future research prospect of this paper. The relevant modifications are as follows.
In addition, this paper investigates the urbanization process from the perspectives of population, land, and economy, which can reflect the comprehensive urbanization level. We believe this has important implications for identifying the sustainability of urbanization. Since there is no authoritative definition of sustainable urbanization, although this paper tries to use multiple urbanization indicators, it still cannot fully reflect sustainable urbanization. In the future, more in-depth research on sustainable urbanization should be carried out based on sustainable development.

(2) The DZ model is the core of the paper, so I suggest looking into the subject in more depth. More information would be appreciated (for example what purpose they serve, through which policies they were introduced and when, how the areas were selected, what criteria were used). (Introduction)
Response: Thank you very much for your suggestion. The "DZ model" is a typical example of government-led urbanization. And the purpose of DZ is to accept the transfer of international capital and industries. According to your suggestion, we have added the relevant expressions of the DZ model, including its establishment purpose, preferential policies, spatial layout, development objectives, development status, etc. The relevant modifications are as follows.
Some scholars have summarized China's urbanization development into seven "promoting models". Among these, the "development zone (DZ) model" is one of the most representative [18]. The " DZ model" is a typical example of government-led urbanization. The government establishes DZ in regions with advantageous locations and provides a series of preferential policies [19]. China's first batch of DZs was all set up in eastern port cities with advantageous locations. To support the development of the DZs, the State gives preferential policies such as tax reduction and financial support. The DZ can accept the transfer of international capital and industries by its policy superiority and favorable investment environment [20]. Since establishing the first national DZ in 1984, DZs have become an important strategic initiative to promote local economic growth [21]. As of 2018, China has established a total of 552 national DZs. The location of DZs has also shifted from the eastern coastal regions to the relatively backward central and western regions. By establishing DZs, cities can complete industrial and population agglomeration [22], and achieve transformation of land structure and industrial structure [23]. Due to the superiority of the "DZ model", it is necessary to estimate the impact of DZs on urbanization.

(3) In row 80 authors wrote: “Therefore, this paper uses the DID model to quantify the role of DZs in China's urbanization to fill the above research gaps”. I suggest making it explicitly clear what gaps are intended to be bridged by the analysis”.
Response: Thank you very much for your suggestion. According to your suggestion, we have summarized three shortcomings of existing research. First, existing studies do not exclude the influence of the external environment. Second, existing studies mostly use a single indicator to analyze the urbanization process, which cannot reflect the level of comprehensive urbanization and sustainable urbanization. Third, current studies have not considered spatial differences in the policy effects of DZs. Therefore, we have made a targeted research design to fill the above gaps. The relevant modifications are as follows.
To sum up, existing studies do not exclude the external environment's influence when analyzing DZ's policy effects of DZs. The policy effects of DZs on urbanization have not been thoroughly examined. Second, existing studies mostly use a single indicator to analyze the urbanization process, which cannot reflect the level of comprehensive urbanization and sustainable urbanization. Third, current studies have not considered spatial differences in the policy effects of DZs. Therefore, this paper uses the DID model to quantify the role of DZs in China's urbanization. We conduct a series of robustness analyses to exclude the influence of external circumstances on the results. Considering that urbanization is a comprehensive process, we evaluate the spillover effects of DZs on urbanization from the perspectives of population, land, and the economy. In addition, this paper identifies the heterogeneous influence of DZs on urbanization to serve targeted urbanization development policies.

(4) In row 89 authors wrote “using the DID method”. The acronym is introduced without giving a definition.
Response: Thank you very much for your reminder. The abstract has defined DID in row 15. According to your reminder, we have added the definition of DID in row 114 to make it easier for readers to understand.

(5) In row 91 authors wrote “from the perspectives of the population, land, and the economy”. It seems a too general sentence, I suggest clarifying which aspects of these elements (population, land, economy) authors intend to investigate.
Response: Thank you very much for your suggestion. The development process of urbanization involves the population, land, economy, and other fields. Therefore, this paper evaluates the spillover effects of DZ on urbanization from the perspectives of the population, land, and the economy. Specifically, when exploring the spillover effects of DZ policy on urbanization, the dependent variables comprehensively consider population urbanization (PU), land urbanization (LU), and economic urbanization (EU). The Variables and Data section has clarified these elements (population, land, economy). Include variable definitions and data sources. The relevant contents are as follows.
Urbanization is not only manifested in the transfer of the rural population to the urban population but is also accompanied by the adjustment of industrial structure and the spread of construction land. The development process of urbanization involves the population, land, economy, and other fields [45, 46]. When exploring the spillover effects of DZ policy, the dependent variables comprehensively consider population urbanization (PU), land urbanization (LU), and economic urbanization (EU). PU is the ratio of the urban population to the total population of a city. LU is the ratio of urban construction land area to the total area. EU is the proportion of the output value of the city's secondary and tertiary industries to its GDP. The population data comes from the "China Regional Statistical Yearbook 1990–2018" The urban construction land data comes from the Environmental Science Data Center of the Chinese Academy of Sciences, and the economic data comes from the "China City Statistical Yearbook 1990–2018".
We think it would be more appropriate to clarify these elements in the Variables and Data section. Therefore, we have deleted the statement in section 2.1 and clarified it in section 2.2. We hope our response can be accepted and welcome your further suggestions.

(6) Please clarify why the DID model was chosen and how it works from a conceptual point of view (for those who are not statisticians).
Response: Thank you very much for your suggestion. According to your suggestion, we have revised the clarification of the DID model. We have added reasons for choosing the DID model. And we also clarify the conceptual point of the DID model so that non-statisticians can better understand the method. The modifications are as follows.
This paper aims to test whether the development zone policy can promote the urbanization process. The difference in difference (DID) model is practical for evaluating policy effects. The model divides the research sample into treatment group (where the policy is implemented) and control group (where the policy is not implemented). Unobservable factors are eliminated by differentiating before and after policy implementation and between the treatment group and the control group, and the policy net effect can be identified. Scholars usually use the DID model to conduct policy evaluation because this model can better avoid the endogenous effects of policies [40-42]. This paper divides the research sample into a treatment group and a control group according to whether the city has already established a DZ. This allows the DID model to be constructed based on the double fixed effects of time and the individual. Because the time of establishing DZs in various cities is not consistent, this article chooses the progressive DID model to test the impact of DZ on urbanization [43, 44]. The expression of the model is as follows:
ln〖PU〗_it=α_0+α_1 〖DZ〗_it+α_2 〖lnZ〗_it+δ_t+γ_i+ε_it,                     (1)
where U_it is the dependent variable, which represents the population urbanization (PU), land urbanization (LU), and economic urbanization (EU) levels of city i at time t, respectively; 〖DZ〗_it is the independent variable, which indicates whether city i has set up a DZ at time t; Z_it represents the control variables such as per capita GDP (PcGDP), average wage (AS), budgetary revenue (BR), fixed asset in-vestment (FAI), foreign direct investment (FDI), political status (PS), nearshore distance (ND), and latitude (LA); α_0, α_1, and α_2 are regression coefficients; δ_t and γ_i are the fixed effects on the time and individual, respectively; and ε_it is a random disturbance term. In order to eliminate the possible heteroscedasticity in the model, the dependent variables and the control variable are processed logarithmically in this paper. In the model regression, the coefficient α_1 is the focus of this paper. The coefficient reflects the impact of the DZ policy on the urbanization process after double difference. If α_1 is significantly positive, it means that the DZ policy can promote the regional urbanization process. 

(7) Why are the 'propensity score matching' and 'kernel matching method' not mentioned in this paragraph on the method?
Response: Thank you very much for your reminder. The purpose of propensity score matching is to test the robustness of DID model. The kernel matching method is one of many matching methods for propensity score matching. Similarly, parallel trend analysis is also a method to test the robustness of DID model. These methods are not the core methods of this paper. Therefore, we think it appropriate to express these methods as a robustness test in the result section. Previous studies often used this presentation. Accordance to your reminder, we have added expressions related to propensity score matching analysis and parallel trend test in the methodology section. The additions are as follows.
The DID model requires that the division of the treatment group and the control group is random. And the treatment group and the control group should also meet the parallel trend assumption before implementing the policy. Therefore, this paper conducts propensity score matching analysis and parallel trend test on the sample to ensure the accuracy of the estimation results. In addition, this paper divides the sample into the eastern, central, and western regions, to identify the differences among policy effects in different regions of China.

(8) Furthermore, measuring social development solely by a purely economic parameter/s does not seem sufficient.
Response: Thank you very much for your reminder. The development of urbanization is not only affected by the DZ policy but also by other factors. Therefore, this paper introduces some control variables into the DID model. These variables are important factors affecting the urbanization process rather than measuring social development. Accordance to your reminder, selecting control variables solely by purely economic parameters does not seem sufficient. Therefore, in the selection of control variables, in addition to economic parameters, we also selected non-economic variables such as political status, nearshore distance, and latitude. We believe that considering both economic and non-economic parameters will make the model more accurate. We hope our response can be accepted and welcome your further suggestions.

(9) There is no critical mention of the fact that the effects of urbanisation should also include issues such as the environment, quality of life, social relations, services, land consumption, climate effects, equity, etc. in their assessment.
Response: Thank you very much for your suggestion. We are entirely in agreement with you that the impact of urbanization involves many aspects. The issues you mentioned, such as the environment, quality of life, social relations, services, land consumption, and climate impact, are also hotspots in urbanization research. According to your suggestion, we believe that the impact mechanism of DZ on urbanization can be discussed from the perspectives of environment, quality of life, social relations, and services in the future. This perspective is interesting and important. Therefore, we have taken it as the future research prospect of this paper.
Another limitation of this paper was that it only focused on the correlation between urbanization and DZ policy without an in-depth discussion on the mechanism. Based on this study, it might prove fruitful to investigate further the impact mechanism of DZs on urbanization from the perspectives of environment, quality of life, social relations, services, and equity.

(10) In row 346 authors wrote: “High-quality urbanization is an important support mechanism for China's sustainable development”. However I think that the parameters used are not adequate to define quality. So what do you mean by “high quality urbanization”?
Response: Thank you very much for your suggestion. We believe that the urbanization process promoted by the DZ is characterized by high quality and sustainability. Because the DZ is the regional industrial agglomeration center and the growth pole of economic development, it has strong development vitality. Urbanization promoted by DZs not only leads to population agglomeration in cities but also changes the land use and industrial structure of cities, which can significantly improve the development quality of urbanization. According to your suggestion, we have decided to delete the expression of high-quality urbanization and replace it with sustainable urbanization. The reasons for using sustainable urbanization have been explained in our response to question 1. We hope the response can be accepted and welcome your further suggestions.

4. Response to Reviewer #4:
This text is a good piece of paper and it gives more insights policy effects of setting up development zones on urbanization from the perspectives of the population, land, and the economy because it evaluates the policies effects. Nevertheless, the paper is a China-based case study that needs more reflection for the international readership. I highly suggest to add a paragraph or a new text in the introduction with the international literature review in the following aspects:
1) the distortions provoked by capitalist urban regeneration processes by taking into references some examples of global cities:
- 2020. Alpha City: How London Was Captured by the Super-Rich. London: Verso
- 2019. Capital City. Gentrification and the real estate state. London-New York: Verso
- 2019. Regenerating Bilbao: From "productive industries" to "productive services". Territorio, 89
2) healthy cities to contrast speculative urbanization:
- 2020. Changing the urban design of cities for health: The superblock model. Environment International, 134: 105132
3) Urbanization and urban sprawl:
https://www.mdpi.com/2071-1050/11/2/485
4) the indirect effects of urbanization in the redevelopment of historic neighborhoods
- 2019. Historic Cities: Issues in Urban Conservation, The Getty Conservation Institute, Los Angeles.
The analysis outstands but needs a more critical literature review. By doing so, the paper will be suitable for publishing and for international readership. Also, write the lessons learned of this analysis in the conclusion. The conclusion is not sufficient for the analysis done by Authors. This is why I require a new (and updated) version of the paper.
Response: Thank you very much for your suggestion. According to your suggestion, we have added the discussion on the urban regeneration process, health of cities and ecological security, urban capital, and land sprawl under the process of urbanization, citing international literature in these fields. The additions are as follows.
The emergence of cities is a sign of human maturity and civilization [1]. After two centuries of unprecedented rapid urbanization, more than half of the world's population lives in cities [2]. Rapid urbanization has a profound impact on global sustainability while increasing social productivity. The research on urbanization began at the beginning of the emergence of modern industrial cities and has been widely concerned by the academic community. Scholars believe that while urbanization promotes social modernization in various countries, it also brings a series of problems. For example, studies in London and New York found that state capital and planning play an essential role in urbanization [3]. In urban renewal, land values and rents have risen rapidly. However, ordinary citizens do not benefit from it, and the the wealth gap problem becomes more significant in urbanization [4]. The rapid development of urbanization has also deepened environmental issues and aroused academic attention to healthy and ecological cities [5]. Scholars have deeply analyzed urban air pollution [6], heat island effect [7], ecological pattern [8], land expansion [9], and other issues under the process of urbanization, and explored sustainable urbanization models.
References:
1. Turok, I.; Mykhnenko, V., The trajectories of European cities, 1960–2005. Cities 2007, 24, (3), 165-182.
2. Mulligan, G. F., Reprint of: Revisiting the urbanization curve. Cities 2013, 32, S58-S67.
3. Stein, S., Capital city: Gentrification and the real estate state. Verso Books: 2019.
4. Atkinson, R., Alpha city: How London was captured by the super-rich. Verso Books: 2021.
5. Mueller, N.; Rojas-Rueda, D.; Khreis, H.; Cirach, M.; Andrés, D.; Ballester, J.; Bartoll, X.; Daher, C.; Deluca, A.; Echave, C., Changing the urban design of cities for health: The superblock model. Environment international 2020, 134, 105132.
6. Todorov, V.; Dimov, I., Innovative digital stochastic methods for multidimensional sensitivity analysis in air pollution modelling. Mathematics 2022, 10, (12), 2146.
7. Piracha, A.; Chaudhary, M. T., Urban air pollution, urban heat island and human health: A review of the literature. Sustainability 2022, 14, (15), 9234.
8. Liu, H.; Li, Q.; Yu, D.; Gu, Y., Air quality index and air pollutant concentration prediction based on machine learning algorithms. Applied Sciences 2019, 9, (19), 4069.
9. Jarah, S. H. A.; Zhou, B.; Abdullah, R. J.; Lu, Y.; Yu, W., Urbanization and urban sprawl issues in city structure: A case of the Sulaymaniah Iraqi Kurdistan Region. Sustainability 2019, 11, (2), 485.
We have also concluded the shortcomings and lessons of the existing research at the end of the introduction. First, existing studies do not exclude the influence of the external environment. Second, current studies mostly use a single indicator to analyze the urbanization process, which cannot reflect the level of comprehensive urbanization and sustainable urbanization. Third, existing studies have not considered spatial differences in the policy effects of DZs. Therefore, we have made a targeted research design to fill the above gaps. The relevant modifications are as follows.
To sum up, existing studies do not exclude the external environment's influence when analyzing DZ's policy effects of DZs. The policy effects of DZs on urbanization have not been thoroughly examined. Second, existing studies mostly use a single indicator to analyze the urbanization process, which cannot reflect the level of comprehensive urbanization and sustainable urbanization. Third, current studies have not considered spatial differences in the policy effects of DZs. Therefore, this paper uses the DID model to quantify the role of DZs in China's urbanization. We conduct a series of robustness analyses to exclude the influence of external circumstances on the results. Considering that urbanization is a comprehensive process, we evaluate the spillover effects of DZs on urbanization from the perspectives of population, land, and the economy. In addition, this paper identifies the heterogeneous influence of DZs on urbanization to serve targeted urbanization development policies.

We have used English editing service to polish the manuscript and provided a polished manuscript as proof of editing. Once again, we appreciate sincerely for the editors/reviewers’ warm and patient work, and thank you very much for your constructive comments and suggestions which really promoted the quality of our manuscript. We hope that our earnest revision will conform to the approval of Land.

*********************************************************************
References
1. Turok, I.; Mykhnenko, V., The trajectories of European cities, 1960–2005. Cities 2007, 24, (3), 165-182.
2. Mulligan, G. F., Reprint of: Revisiting the urbanization curve. Cities 2013, 32, S58-S67.
3. Stein, S., Capital city: Gentrification and the real estate state. Verso Books: 2019.
4. Atkinson, R., Alpha city: How London was captured by the super-rich. Verso Books: 2021.
5. Mueller, N.; Rojas-Rueda, D.; Khreis, H.; Cirach, M.; Andrés, D.; Ballester, J.; Bartoll, X.; Daher, C.; Deluca, A.; Echave, C., Changing the urban design of cities for health: The superblock model. Environment international 2020, 134, 105132.
6. Todorov, V.; Dimov, I., Innovative digital stochastic methods for multidimensional sensitivity analysis in air pollution modelling. Mathematics 2022, 10, (12), 2146.
7. Piracha, A.; Chaudhary, M. T., Urban air pollution, urban heat island and human health: A review of the literature. Sustainability 2022, 14, (15), 9234.
8. Liu, H.; Li, Q.; Yu, D.; Gu, Y., Air quality index and air pollutant concentration prediction based on machine learning algorithms. Applied Sciences 2019, 9, (19), 4069.
9. Jarah, S. H. A.; Zhou, B.; Abdullah, R. J.; Lu, Y.; Yu, W., Urbanization and urban sprawl issues in city structure: A case of the Sulaymaniah Iraqi Kurdistan Region. Sustainability 2019, 11, (2), 485.
10. Chen, M.; Liu, W.; Tao, X., Evolution and assessment on China's urbanization 1960–2010: under-urbanization or over-urbanization? Habitat International 2013, 38, 25-33.
11. Chen, M.; Liu, W.; Lu, D.; Chen, H.; Ye, C., Progress of China's new-type urbanization construction since 2014: A preliminary assessment. Cities 2018, 78, 180-193.
12. Wang, X.-R.; Hui, E. C.-M.; Choguill, C.; Jia, S.-H., The new urbanization policy in China: Which way forward? Habitat International 2015, 47, 279-284.
13. Bai, X.; Shi, P.; Liu, Y., Society: Realizing China's urban dream. Nature 2014, 509, (7499), 158-160.
14. Chen, M.; Zhou, Y.; Huang, X.; Ye, C., The integration of new-type urbanization and rural revitalization strategies in China: origin, reality and future trends. Land 2021, 10, (2), 207.
15. Guan, X.; Wei, H.; Lu, S.; Dai, Q.; Su, H., Assessment on the urbanization strategy in China: Achievements, challenges and reflections. Habitat International 2018, 71, 97-109.
16. Feng, W.; Liu, Y.; Qu, L., Effect of land-centered urbanization on rural development: A regional analysis in China. Land Use Policy 2019, 87, 104072.
17. Li, Y.; Li, Y.; Zhou, Y.; Shi, Y.; Zhu, X., Investigation of a coupling model of coordination between urbanization and the environment. Journal of environmental management 2012, 98, 127-133.
18. Li, Q.; Chen, Y.; Liu, J., On the development mode of Chinese urbanization. Social Sciences in China 2012, 7, 82-100.
19. Gao, S.; Wang, S.; Sun, D., Development zones and their surrounding host cities in China: Isolation and mutually beneficial interactions. Land 2021, 11, (1), 20.
20. Chen, B.; Lu, M.; Timmins, C.; Xiang, K. Spatial misallocation: Evaluating place-based policies using a natural experiment in China; National Bureau of Economic Research: 2019.
21. Wei, Y. D.; Leung, C. K., Development zones, foreign investment, and global city formation in Shanghai. Growth and Change 2005, 36, (1), 16-40.
22. Cheng, H.; Liu, Y.; He, S.; Shaw, D., From development zones to edge urban areas in China: A case study of Nansha, Guangzhou City. Cities 2017, 71, 110-122.
23. Jiang, Y.; Waley, P., Keeping up with the zones (es): how competing local governments in China use development zones as back doors to urbanization. Urban Geography 2022, 1-21.
24. Schwarz, J. E.; Volgy, T. J., Experiments in employment-A British cure. Harvard Business Review 1988, 66, (2), 104-112.
25. Erickson, R. A.; Syms, P. M., The effects of enterprise zones on local property markets. Regional Studies 1986, 20, (1), 1-14.
26. Papke, L. E., Tax policy and urban development: evidence from the Indiana enterprise zone program. Journal of Public Economics 1994, 54, (1), 37-49.
27. Bondonio, D.; Greenbaum, R. T., Do local tax incentives affect economic growth? What mean impacts miss in the analysis of enterprise zone policies. Regional science and urban economics 2007, 37, (1), 121-136.
28. Greenbaum, R. T.; Engberg, J. B., The impact of state enterprise zones on urban manufacturing establishments. Journal of Policy Analysis and Management 2004, 23, (2), 315-339.
29. Neumark, D.; Kolko, J., Do enterprise zones create jobs? Evidence from California’s enterprise zone program. Journal of Urban Economics 2010, 68, (1), 1-19.
30. Begg, I., High technology location and the urban areas of Great Britain: developments in the 1980s. Urban studies 1991, 28, (6), 961-981.
31. Gobillon, L.; Magnac, T.; Selod, H., Do unemployed workers benefit from enterprise zones? The French experience. Journal of Public Economics 2012, 96, (9-10), 881-892.
32. Li, Y.; Wang, X., Innovation in suburban development zones: Evidence from Nanjing, China. Growth and Change 2019, 50, (1), 114-129.
33. Wang, J., The economic impact of special economic zones: Evidence from Chinese municipalities. Journal of development economics 2013, 101, 133-147.
34. Lu, Y.; Wang, J.; Zhu, L., Do place-based policies work? Micro-Level evidence from China's economic zone program. Micro-Level Evidence from China's Economic Zone Program (July 3, 2015) 2015.
35. Sun, Y.; Ma, A.; Su, H.; Su, S.; Chen, F.; Wang, W.; Weng, M., Does the establishment of development zones really improve industrial land use efficiency? Implications for China’s high-quality development policy. Land Use Policy 2020, 90, 104265.
36. Huang, Z.; He, C.; Wei, Y. D., A comparative study of land efficiency of electronics firms located within and outside development zones in Shanghai. Habitat International 2016, 56, 63-73.
37. Wong, S.-W.; Tang, B.-s., Challenges to the sustainability of ‘development zones’: A case study of Guangzhou Development District, China. Cities 2005, 22, (4), 303-316.
38. Cartier, C., 'Zone fever', the arable land debate, and real estate speculation: China's evolving land use regime and its geographical contradictions. Journal of Contemporary China 2001, 10, (28), 445-469.
39. Zhang, J., Interjurisdictional competition for FDI: The case of China's “development zone fever”. Regional Science and Urban Economics 2011, 41, (2), 145-159.
40. Cui, J.; Zhang, J.; Zheng, Y. In Carbon pricing induces innovation: evidence from China's regional carbon market pilots, AEA Papers and Proceedings, 2018; 2018; pp 453-57.
41. Heckert, M.; Mennis, J., The economic impact of greening urban vacant land: a spatial difference-in-differences analysis. Environment and Planning A 2012, 44, (12), 3010-3027.
42. Sivadasan, J., Barriers to competition and productivity: Evidence from India. The BE Journal of Economic Analysis & Policy 2009, 9, (1).
43. Beck, T.; Levine, R.; Levkov, A., Big bad banks? The winners and losers from bank deregulation in the United States. The Journal of Finance 2010, 65, (5), 1637-1667.
44. Wing, C.; Simon, K.; Bello-Gomez, R. A., Designing difference in difference studies: best practices for public health policy research. Annu Rev Public Health 2018, 39, (1), 453-469.
45. Yu, B., Ecological effects of new-type urbanization in China. Renewable and Sustainable Energy Reviews 2021, 135, 110239.
46. Lv, T.; Wang, L.; Zhang, X.; Xie, H.; Lu, H.; Li, H.; Liu, W.; Zhang, Y., Coupling coordinated development and exploring its influencing factors in Nanchang, China: From the perspectives of land urbanization and population urbanization. Land 2019, 8, (12), 178.
47. Deng, X.; Huang, J.; Rozelle, S.; Zhang, J.; Li, Z., Impact of urbanization on cultivated land changes in China. Land use policy 2015, 45, 1-7.
48. Zhang, H.; Chen, M.; Liang, C., Urbanization of county in China: Spatial patterns and influencing factors. Journal of Geographical Sciences 2022, 32, (7), 1241-1260.
49. Chen, A.; Partridge, M. D., When are cities engines of growth in China? Spread and backwash effects across the urban hierarchy. Regional Studies 2013, 47, (8), 1313-1331.
50. Kuang, W.; Liu, J.; Dong, J.; Chi, W.; Zhang, C., The rapid and massive urban and industrial land expansions in China between 1990 and 2010: A CLUD-based analysis of their trajectories, patterns, and drivers. Landscape and Urban Planning 2016, 145, 21-33.
51. He, C.; Zhao, Y.; Huang, Q.; Zhang, Q.; Zhang, D., Alternative future analysis for assessing the potential impact of climate change on urban landscape dynamics. Science of the Total Environment 2015, 532, 48-60.
52. Gebel, M.; Voßemer, J., The impact of employment transitions on health in Germany. A difference-in-differences propensity score matching approach. Social science & medicine 2014, 108, 128-136.
53. Blundell, R.; Costa Dias, M., Evaluation methods for non‐experimental data. Fiscal studies 2000, 21, (4), 427-468.
54. Rosenbaum, P. R.; Rubin, D. B., The central role of the propensity score in observational studies for causal effects. Biometrika 1983, 70, (1), 41-55.
55. Gao, S.; Zhang, J.; Mo, X.; Wu, R., Dynamic Evolution of the Operating Efficiency of Development Zones in China. Sustainability 2021, 13, (18), 10395.

Reviewer 4 Report

This text is a good piece of paper and it gives more insights policy effects of setting up development zones on urbanization from the perspectives of the population, land, and the economy because it evaluates the policies effects.

Nevertheless, the paper is a China-based case study that needs more reflection for the international readership. 

I highly suggest to add a paragraph or a new text in the introduction with the international literature review in the following aspects: 

1) the distortions provoked by capitalist urban regeneration processes by taking into references some examples of global cities:

- 2020. Alpha City: How London Was Captured by the Super-Rich. London: Verso

- 2019. Capital City. Gentrification and the real estate state. London-New York: Verso

- 2019. Regenerating Bilbao: From "productive industries" to "productive services". Territorio, 89

2) healthy cities to contrast speculative urbanization:

- 2020. Changing the urban design of cities for health: The superblock model. Environment International, 134: 105132

3) Urbanization and urban sprawl:

https://www.mdpi.com/2071-1050/11/2/485

4) the indirect effects of urbanization in the redevelopment of historic neighborhoods

- 2019. Historic Cities: Issues in Urban Conservation, The Getty Conservation Institute, Los Angeles.

The analysis outstands but needs a more critical literature review. By doing so, the paper will be suitable for publishing and for international readership. Also, write the lessons learned of this analysis in the conclusion. The conclusion is not sufficient for the analysis done by Authors.

This is why I require a new (and updated) version of the paper.

Author Response

(The authors gave the same response as above.)

Round 2

Reviewer 4 Report

Dear all,

the new version is good but all the references suggested were not added so I suggest to do so.

The choice is on the authors, it is djudt a suggestion